# Computing Optimal Transport Maps and Wasserstein Barycenters Using Conditional Normalizing Flows

**Gabriele Visentin** [1] **Patrick Cheridito** [1]

## Abstract

We present a novel method for efficiently computing optimal transport maps and Wasserstein barycenters in high-dimensional spaces. Our approach uses conditional normalizing flows to approximate the input distributions as invertible pushforward transformations from a common latent space. This makes it possible to directly solve the primal problem using gradient-based minimization of the transport cost, unlike previous methods that rely on dual formulations and complex adversarial optimization. We show how this approach can be extended to compute Wasserstein barycenters by solving a conditional variance minimization problem. A key advantage of our conditional architecture is that it enables the computation of barycenters for hundreds of input distributions, which was computationally infeasible with previous methods. Our numerical experiments illustrate that our approach yields accurate results across various high-dimensional tasks and compares favorably with previous state-of-the-art methods.

## 1. Introduction

Optimal transport (OT) and Wasserstein barycenters are active research areas with applications in economics, operations research, statistics, physics, PDE theory, and machine learning (Peyré & Cuturi, 2019; Villani, 2021). OT has been successfully applied to generative modeling (Arjovsky et al., 2017; Petzka et al., 2017; Wu et al., 2018; Liu et al., 2019; Cao et al., 2019; Leygonie et al., 2019) and domain adaptation (Luo et al., 2018; Shen et al., 2018; Xie et al., 2019), while Wasserstein barycenters, by providing a natural notion of average for probability distributions, found

applications in problems of shape interpolation (Solomon et al., 2015), image interpolation (Lacombe et al., 2023; Simon & Aberdam, 2020), color translation (Rabin et al., 2014), style translation (Mroueh, 2019), Bayesian subset posterior estimation (Srivastava et al., 2015), clustering in Wasserstein space (Del Barrio et al., 2019; Ho et al., 2017), and fairness (Chzhen et al., 2020; Gouic et al., 2020).

Traditional numerical methods work with discrete or discretized probability measures (Peyré & Cuturi, 2019), via linear programming (Anderes et al., 2016) and entropy regularization (Cuturi & Doucet, 2014; Solomon et al., 2015). They yield accurate results for discrete measures on low-dimensional spaces but scale poorly in the number of support points, so that their performance degrades for continuous distributions, especially in high-dimensional settings (Fan et al., 2020; Korotin et al., 2021b).

These limitations have generated substantial interest in numerical methods for OT and Wasserstein barycenters problems for continuous distributions. While other methods are reviewed in Section 4, we note here that almost all of them solve the Kantorovich dual of the OT problem and employ neural networks to parametrize either the dual potentials (Li et al., 2020) or the transport maps (Kolesov et al., 2024), which often results in complex bi-level (Kolesov et al., 2024; Korotin et al., 2019; Xie et al., 2019; Lu et al., 2020) or tri-level (Fan et al., 2020) adversarial learning.

**Contributions.** We propose a different approach. Our main contributions are the following:

1. We propose a method that bypasses complex adversarial training by directly solving the primal formulation of the OT problem. This approach directly yields both OT maps and values. In addition, it enables an intuitive reformulation of the Wasserstein barycenter problem as an expected conditional variance minimization problem.

2. In our setup, OT maps are bijective. Therefore, we approximate them with conditional normalizing flows, which are flexible bijective models and therefore a natural choice in this context.

3. For barycenter problems, our method directly yields

[1]Department of Mathematics, ETH Zurich, Switzerland. Correspondence to: Gabriele Visentin <gabriele.visentin@math.ethz.ch>.

*Proceedings of the $42^{nd}$ International Conference on Machine Learning*, Vancouver, Canada. PMLR 267, 2025. Copyright 2025 by the author(s).

OT maps and their inverses between the input distributions and the barycenter without requiring additional computations. Moreover, it provides a generative model of the barycenter which allows for direct sampling without querying the input distributions.

4. Our model's conditional architecture scales well in the number of input distributions and can be used to compute Wasserstein barycenters of hundreds of input distributions, which is computationally infeasible with existing methods.

## 2. Background and Preliminaries

In this section we recall the OT problem (Section 2.1) and the Wasserstein barycenter problem (Section 2.2).

**Notation.** We denote by $\|\cdot\|$ the Euclidean norm on $\mathbb{R}^d$ and by $L^p(\mu)$ the space $L^p(\mathbb{R}^d, \mathcal{B}(\mathbb{R}^d), \mu)$ equipped with the norm $\|f\|_{L^p(\mu)} := \left(\int_{\mathbb{R}^d} \|f(z)\|^p \mu(dz)\right)^{1/p}$. $\mathcal{P}(\mathbb{R}^d)$ denotes the space of all Borel probability measures on $\mathbb{R}^d$, $\mathcal{P}_p(\mathbb{R}^d)$ is the space of all $\mu \in \mathcal{P}(\mathbb{R}^d)$ such that $\int_{\mathbb{R}^d} \|x\|^p \mu(dx) < \infty$, while $\mathcal{P}_{ac}(\mathbb{R}^d)$ is the space of all $\mu \in \mathcal{P}(\mathbb{R}^d)$ that are absolutely continuous with respect to the Lebesgue measure. Finally, we define $\mathcal{P}_{p,ac}(\mathbb{R}^d) := \mathcal{P}_p(\mathbb{R}^d) \cap \mathcal{P}_{ac}(\mathbb{R}^d)$.

### 2.1. Optimal Transport

The Monge-formulation (Monge, 1781) of the OT problem between two distributions $\mu, \nu \in \mathcal{P}(\mathbb{R}^d)$ for $p$-cost is:

$$\mathbb{W}_p^p(\mu, \nu) = \inf_{T_\#\mu=\nu} \int_{\mathbb{R}^d} \|x - T(x)\|^p \mu(dx), \qquad (1)$$

where $T_\#\mu$ denotes the pushforward measure of $\mu$ under $T$. This problem is difficult to handle and does not always have a solution. Therefore, it was relaxed by Kantorovich (1942) as follows:

$$\mathbb{W}_p^p(\mu, \nu) = \inf_{\gamma \in \Gamma(\mu, \nu)} \int_{\mathbb{R}^d} \|x - y\|^p \gamma(dx, dy), \qquad (2)$$

where $\Gamma(\mu, \nu)$ is the set of all couplings between $\mu$ and $\nu$, i.e. the set of all probability measures in $\mathcal{P}(\mathbb{R}^d \times \mathbb{R}^d)$ with marginals $\mu$ and $\nu$.

However, under some mild regularity assumptions, problems (1) and (2) are equivalent (see Lemma 3.1 below), and $\mathbb{W}_p(\mu, \nu)$ defines a metric on the space $\mathcal{P}_p(\mathbb{R}^d)$, known as the Wasserstein-$p$ metric.

### 2.2. Wasserstein Barycenters

Wasserstein barycenters are Fréchet means on the Wasserstein spaces $(\mathcal{P}_p(\mathbb{R}^d), \mathbb{W}_p)$. In the following we focus on the case $p = 2$, which is of most practical interest. Let $(\mu_s)_{s \in \mathcal{S}} \subseteq \mathcal{P}_{2,ac}(\mathbb{R}^d)$ be a family of distributions on $\mathbb{R}^d$ indexed by a finite set $\mathcal{S}$ and $(w_s, s \in \mathcal{S})$ non-negative weights such that $\sum_{s \in \mathcal{S}} w_s = 1$. Then the $w$-weighted Wasserstein-2 barycenter is given by:

$$\bar{\mu} = \arg\inf_{\nu \in \mathcal{P}_2(\mathbb{R}^d)} \sum_{s \in \mathcal{S}} w_s \mathbb{W}_2^2(\nu, \mu_s). \qquad (3)$$

Under these assumptions, the Wasserstein-2 barycenter $\bar{\mu}$ exists, is unique and is absolutely continuous with respect to the Lebesgue measure (Agueh & Carlier, 2011; Brizzi et al., 2025).

## 3. Proposed Method

In this section, we first introduce some additional notation (Section 3.1). Then we derive our main theoretical results (Section 3.2) and present their algorithmic implementation (Section 3.3). The proofs of all lemmas and theorems can be found in Appendix B.

### 3.1. Notation

Given two Borel probability measures $\mu, \nu \in \mathcal{P}(\mathbb{R}^d)$ and a Borel function $f : \mathbb{R}^d \to \mathbb{R}^d$, we say that a Borel function $\tilde{f} : \mathbb{R}^d \to \mathbb{R}^d$ is a $(\mu, \nu)$-inverse of $f$ if $\tilde{f} \circ f(x) = x$ for $\mu$-almost all $x \in \mathbb{R}^d$ and $f \circ \tilde{f}(x) = x$ for $\nu$-almost all $x \in \mathbb{R}^d$.

We denote by $B(\mu, \nu)$ the set of all Borel functions $f : \mathbb{R}^d \to \mathbb{R}^d$ such that $f_\#\mu = \nu$ and $f$ admits a $(\mu, \nu)$-inverse. In Lemmas A.3 and A.4 below we show some basic facts about this set. In particular, we show that if $\mu, \nu \in \mathcal{P}_p(\mathbb{R}^d)$ and $f \in B(\mu, \nu)$, then $\|f\|$ belongs to $L^p(\mu)$, the $(\mu, \nu)$-inverse $\tilde{f}$ is $\nu$-almost surely unique and it belongs to $B(\nu, \mu)$. A special case is when $\mu$ and $\nu$ have non-vanishing densities with respect to the Lebesgue measure on $\mathbb{R}^d$, then $B(\mu, \nu)$ is just the set of all almost everywhere bijective pushforward maps that push $\mu$ to $\nu$. Furthermore, if $f \in B(\mu, \nu)$, then its $(\mu, \nu)$-inverse is almost everywhere unique and coincides almost everywhere with the inverse map $f^{-1}$, whenever the latter exists.

### 3.2. Theoretical Results

Bijective pushforward maps arise naturally in OT theory, as shown in the following version of Brenier's theorem, which follows from Gangbo & McCann (1996).

**Lemma 3.1.** *Let $p > 1$ and $\mu, \nu \in \mathcal{P}_{p,ac}(\mathbb{R}^d)$. Then, Kantorovich's problem* (2) *has a unique solution $\gamma \in \Gamma(\mu, \nu)$. Moreover, $\gamma$ is of the form $(id, T)_\#\mu$, where $T \in B(\mu, \nu)$ is a $\mu$-almost surely unique solution to Monge's problem* (1).

The next theorem allows us to reformulate the OT problem

with $p$-cost as a constrained $L^p(\lambda)$-minimization problem for a given latent distribution $\lambda \in \mathcal{P}_{p,ac}(\mathbb{R}^d)$.

**Theorem 3.2.** *Let* $\lambda, \mu, \nu \in \mathcal{P}_{p,ac}(\mathbb{R}^d)$ *for some* $p > 1$. *Then:*

$$\mathbb{W}_p^p(\mu, \nu) = \min_{\substack{f \in B(\lambda, \mu) \\ g \in B(\lambda, \nu)}} \|f - g\|_{L^p(\lambda)}^p.$$

*Furthermore, if* $f$ *and* $g$ *are solutions of the optimization problem above and* $\tilde{f}$ *is a* $(\lambda, \mu)$-*inverse of* $f$, *then* $g \circ \tilde{f}$ *is an optimal transport map between* $\mu$ *and* $\nu$.

Theorem 3.2 can be understood as a generalization of the closed-form solution of OT in one dimension using quantile functions. Recall, that for $\mu, \nu \in \mathcal{P}_p(\mathbb{R})$, one has:

$$\mathbb{W}_p^p(\mu, \nu) = \|q_\mu - q_\nu\|_{L^p([0,1])}^p,$$

where $L^p([0,1])$ is the $L^p$ space over $[0,1]$ equipped with the uniform distribution and $q_\mu$ is the quantile function of $\mu$ (see, for instance, Remark 2.30 in Peyré & Cuturi (2019)). Theorem 3.2 shows that for $\mu, \nu \in \mathcal{P}_{p,ac}(\mathbb{R}^d)$ we can replace the uniform distribution with any absolutely continuous latent distribution $\lambda$ and the comonotonicity constraint on $q_\mu$ and $q_\nu$ (which is implicit in the definition of quantile function) with closeness in $L^p(\lambda)$.

Our next result shows how to compute Wasserstein-2 barycenters as weighted averages of certain bijective pushforward maps that minimize an expected conditional variance criterion. Consider a non-empty finite set $\mathcal{S} = \{1, \ldots, n\}$ and a collection of probability distributions $\mu_s \in \mathcal{P}_{2,ac}(\mathbb{R}^d)$, $s \in \mathcal{S}$, together with weights $w_s \geq 0$, $s \in \mathcal{S}$, such that $\sum_{s \in \mathcal{S}} w_s = 1$. Then the $w$-weighted Wasserstein-2 barycenter of $(\mu_s)_{s \in \mathcal{S}}$ is the unique $\bar{\mu} \in \mathcal{P}_{2,ac}(\mathbb{R}^d)$ satisfying Equation (3). In the following theorem, we realize $\bar{\mu}$ as the distribution of a $d$-dimensional random vector of the form $h(Z)$ for a Borel function $h : \mathbb{R}^d \to \mathbb{R}^d$ and a $d$-dimensional random vector $Z$ defined on an underlying probability space $(\Omega, \mathcal{F}, \mathbb{P})$ that also carries an independent $\mathcal{S}$-valued random variable $S$ such that $\mathbb{P}(S = s) = w_s$ for all $s \in \mathcal{S}$. The expectation and variance corresponding to $\mathbb{P}$ are denoted by $\mathbb{E}$ and $\text{Var}$, respectively.

**Theorem 3.3.** *Let* $\mathcal{S}$ *be a non-empty finite set and consider a collection of probability distributions* $\mu_s \in \mathcal{P}_{2,ac}(\mathbb{R}^d)$, $s \in \mathcal{S}$, *and weights* $w_s \geq 0$, $s \in \mathcal{S}$, *satisfying* $\sum_{s \in \mathcal{S}} w_s = 1$. *Let* $Z$ *be a* $d$-*dimensional random vector with an arbitrary distribution* $\lambda \in \mathcal{P}_{2,ac}(\mathbb{R}^d)$ *and* $S$ *an independent* $\mathcal{S}$-*valued random variable taking the values* $s$ *with probabilities* $w_s$, $s \in \mathcal{S}$. *Then, the following hold:*

1. *The minimization problem*

$$\min_{\substack{f:\mathbb{R}^d \times \mathcal{S} \to \mathbb{R}^d \\ s.t.\ f(\cdot, s) \in B(\lambda, \mu_s) \\ \text{for all } s \in \mathcal{S}}} \sum_{i=1}^d \mathbb{E}\left[\text{Var}\left(f_i(Z, S) \mid Z\right)\right] \quad (4)$$

*has a solution, and*

2. *For every solution* $f : \mathbb{R}^d \times \mathcal{S} \to \mathbb{R}^d$ *of problem* (4), *the* $w$-*weighted Wasserstein-2 barycenter of* $(\mu_s)_{s \in \mathcal{S}}$ *is equal to the distribution of* $h(Z)$, *where* $h : \mathbb{R}^d \to \mathbb{R}^d$ *is given by the weighted sum*

$$h(z) = \sum_{s \in \mathcal{S}} w_s f(z, s).$$

## 3.3. Implementation

Our approach uses conditional normalizing flows. After introducing them in Section 3.3.1, we present our algorithms for OT maps (Section 3.3.2) and Wasserstein barycenters (Section 3.3.3). In Appendix C we provide all necessary background information on normalizing flows.

### 3.3.1. CONDITIONAL NORMALIZING FLOWS.

Given a finite set of input distributions $(\mu_s)_{s \in \mathcal{S}} \in \mathcal{P}_{p,ac}(\mathbb{R}^d)$ and a reference distribution $\lambda \in \mathcal{P}_{p,ac}(\mathbb{R}^d)$, we use *conditional normalizing flows* to parametrize a map $f : \mathbb{R}^d \times \mathcal{S} \to \mathbb{R}^d$ such that $f(\cdot, s) \in B(\lambda, \mu_s)$ for all $s \in \mathcal{S}$, and we use this parametrization to solve the optimization problems in Theorem 3.2 and Theorem 3.3 with gradient descent in parameter space. Teshima et al. (2020) have shown that coupling-based normalizing flows are universal approximators for bijections with the caveat that their construction relies on ill-conditioned networks (Koehler et al., 2021; Draxler et al., 2024). On the other hand, Draxler et al. (2024) have introduced a well-conditioned flow which can approximate arbitrary distributions but not necessarily every bijection.

We choose the standard $d$-dimensional Gaussian distribution as latent distribution $\lambda$ and we enforce the pushforward constraint $f(\cdot, s)_\# \lambda = \mu_s$ via conditional likelihood maximization. The conditioning variable $s \in \mathcal{S}$ may be $\mathbb{R}$-valued, taking finitely many values (as in Section 5.2.4, where $\mathcal{S}$ is a finite grid in $[0, 1]$) or one-hot encoded (as in all other numerical experiments), depending on how regular we expect the dependence of $\mu_s$ on $s$ to be.

We design and implement our own conditional versions of Real NVP (Dinh et al., 2016) and Glow (Kingma & Dhariwal, 2018) with multi-scale architecture. In the case of Real NVP, we use a conditional affine coupling layer with the following coupling function:

$$f_{\theta(z^B, s)}(z^A) = z^A \odot \exp(a(z^B, s)) + b(z^B, s),$$

where the parameters $\theta(\cdot, \cdot) = (a(\cdot, \cdot), b(\cdot, \cdot))$ are feedforward neural networks, with an additional residual connection for the conditioning input $s$. A similar conditional architecture, but without the residual connection, was already proposed by Atanov et al. (2019), but in our experiments,

we found that adding this residual connection improves the performance. In the case of Glow, we obtain a conditional architecture simply by concatenating the conditioning variable $s$ pixel-wise with the latent input image.

### 3.3.2. COMPUTING OT MAPS

Our method for computing OT maps using Theorem 3.2 is presented in Algorithm 1. The algorithm is given for the case of quadratic cost (i.e. $p = 2$), but it can also be used for OT costs of the form $c(x, y) = h(x - y)$ for a strictly convex function $h \colon \mathbb{R}^d \to [0, \infty)$ satisfying the conditions (H1)–(H3) of Gangbo & McCann (1996).

As shown in Algorithm 1, we learn the source distribution $\mu = \mu_1$ and the target distribution $\nu = \mu_2$ using a joint conditional normalizing flow with conditioning variable $s \in \{1, 2\}$. In principle, it would be possible to use two separate normalizing flows for $\mu_1$ and $\mu_2$. But in our experiments, a conditional model resulted in better performance, probably due to the fact that this architecture introduces a smooth dependence on the conditioning variable $s \in \{0, 1\}$, which facilitates learning pushforward maps that are close in $L^2(\lambda)$.

---

**Algorithm 1** OT Map via Conditional Normalizing Flows.

---

**Input:** input distributions $\mu_1$ and $\mu_2$ accessible by sampling; conditional normalizing flow $f_\theta(\cdot, \cdot) : \mathbb{R}^d \times \{1, 2\} \to \mathbb{R}^d$ with initialized parameters $\theta$ and latent distribution $\lambda$; number of iterations $T$; learning rate $\eta$; decreasing weights $(\zeta_t)_{t=1}^T$.

**Output:** OT map from $\mu_1$ to $\mu_2$ given by $x \mapsto f_\theta(f_\theta^{-1}(x, 1), 2)$.

**for** $t = 1$ **to** $T$ **do**

    sample $S \sim \mathrm{Uniform}(\{1, 2\})$, $X \sim \mu_S$, $Z \sim \lambda$;

    compute model likelihood on $(X, S)$:

    $p_\theta(X, S) = p_Z(f_\theta^{-1}(X, S)) \cdot |\det(Df_\theta^{-1}(X, S))|$;

    compute $L^2$-cost:

    $L_\theta^2(Z) = \|f_\theta(Z, 1) - f_\theta(Z, 2)\|^2$;

    update model parameters $\theta$ by gradient descent:

    $\theta \leftarrow \theta - \eta \nabla_\theta \left(-\log(p_\theta(X, S)) + \zeta_t L_\theta^2(Z)\right)$.

**end for**

---

**Decreasing weights.** Both Algorithm 1 and Algorithm 2 require as input a sequence of decreasing weights $(\zeta_t)_{t=1}^T$ for the $L^2$-cost. This is motivated by the fact that we are trying to solve a multi-objective optimization problem with two competing objectives: the pushforward maps need to be as close as possible in $L^2(\lambda)$ while still being far enough to satisfy their pushforward constraints. Naively minimizing both the $L^2$-cost and the negative log-likelihood in each gradient step leads to poor performance. Instead, we gradually decrease the importance of the $L^2$-cost, thus guaranteeing that at convergence the pushforward constraints are satisfied. The optimal choice of weights depends on the relative

values of the optimal likelihood loss and $L^2$-cost. In all our numerical experiments we chose an exponentially decreasing schedule; see Appendix D for a full description of the hyperparameters used in each numerical experiment.

At the end of training, the OT map from $\mu_1$ to $\mu_2$ can be efficiently retrieved by performing an inverse pass from $\mu_1$ to $\lambda$ followed by a forward pass from $\lambda$ to $\mu_2$, i.e. it is given by $x \mapsto f_\theta(f_\theta^{-1}(x, 1), 2)$.

### 3.3.3. COMPUTING WASSERSTEIN-2 BARYCENTERS.

The algorithm for computing Wasserstein-2 barycenters builds upon Theorem 3.3 and is presented in Algorithm 2.

---

**Algorithm 2** Wasserstein-2 Barycenter via Conditional Normalizing Flows.

---

**Input:** input distributions $(\mu_s)_{s \in \mathcal{S}}$ accessible by sampling; weights $(w_s)_{s \in \mathcal{S}}$ and weighting measure $\sigma(s) := w_s$; conditional normalizing flow $f_\theta(\cdot, \cdot) : \mathbb{R}^d \times \mathcal{S} \to \mathbb{R}^d$ with initialized parameters $\theta$ and latent distribution $\lambda$; number of iterations $T$; learning rate $\eta$; decreasing weights $(\zeta_t)_{t=1}^T$.

**Output:** learned Wasserstein-2 barycenter $h_\#\lambda \approx \bar{\mu}$, where $h(z) = \sum_{s \in \mathcal{S}} w_s f_\theta(z, s)$.

**for** $t = 1$ **to** $T$ **do**

    sample $S \sim \sigma$, $X \sim \mu_S$, $Z \sim \lambda$;

    compute model likelihood on $(X, S)$:

    $p_\theta(X, S) = p_Z(f_\theta^{-1}(X, S)) \cdot |\det(Df_\theta^{-1}(X, S))|$;

    compute $L^2$-cost:

    $L_\theta^2(Z, S) = \|f_\theta(Z, S) - \sum_{s \in \mathcal{S}} w_s f_\theta(Z, s)\|^2$;

    update model parameters $\theta$ by gradient descent:

    $\theta \leftarrow \theta - \eta \nabla_\theta \left(-\log(p_\theta(X, S)) + \zeta_t L_\theta^2(Z, S)\right)$.

**end for**

---

Also in this case we employ a conditional normalizing flow architecture and train using a sequence of decreasing weights. At the end of training, we obtain a generative model of the barycenter given by $h_\#\lambda$ for the function $h(z) = \sum_{s \in \mathcal{S}} w_s f_\theta(z, s)$. Additionally, the OT map from any input distribution $\mu_s$ to the barycenter is given by $x \mapsto h(f_\theta^{-1}(x, s))$, while the OT map from a sampled barycenter datapoint $x = h(z)$ to an input distribution $\mu_s$ is given by $x \mapsto f_\theta(h^{-1}(x), s) = f_\theta(z, s)$.

## 4. Related Work

Traditional numerical methods tackle OT between discrete or discretized distributions with linear programming (Anderes et al., 2016; Peyré & Cuturi, 2019). The Sinkhorn–Knopp algorithm (Sinkhorn & Knopp, 1967; Cuturi, 2013) yields good results for entropy-regularized discrete OT problems in low dimensions. But discrete methods loose accuracy for continuous distributions and become infeasible in high-dimensional settings. This has lead to continuous

methods, most of which tackle a dual formulation of the OT problem with neural networks or kernel expansions; see e.g.Korotin et al. (2021a).

Methods for Wasserstein barycenters can be divided into non-generative and generative. While non-generative models are limited to transporting existing samples from the input distributions, generative models output a learned barycenter distribution which can be used for sampling.

**Non-generative models.** An early example of such a model has been given by Li et al. (2020), who exploit the dual formulation of the entropic regularized Wasserstein barycenter problem and paramatrize the potentials using neural networks. Their method produces only approximate barycenters, due to the bias introduced by the entropic regularization, and outputs only the learned potentials, instead of the OT maps. Korotin et al. (2021b) parametrize potentials with input-convex neural networks (ICNNs) (Amos et al., 2017) and tackle the dual OT problem by imposing a congruence and a cycle-consistency condition. The congruence condition requires fixing a dominating measure that may be difficult to choose *a priori*, while there is evidence that the cycle-consistency condition (Amos et al., 2017) and the use of ICNNs (Korotin et al., 2021a) may be suboptimal. Kolesov et al. (2024) present a model that can be applied to generic cost functionals (both regularized and exact), but relies on bi-level adversarial learning and does not provide a generative model of the barycenter.

**Generative models.** More closely related to our approach are generative models of Wasserstein barycenters. Also for this class of models the dominant approach is to start from the dual formulation and to solve a tri-level or bi-level optimization problem. Fan et al. (2020) solve the barycenter problem by computing Wasserstein distances through a min-max-min tri-level optimization problem using ICNNs, which may be unstable and may result in under-training of the generative model of the barycenter (Korotin et al., 2021b). Korotin et al. (2022) use the fixed point algorithm of Álvarez-Esteban et al. (2016) and the reversed maximin neural OT solver by Dam et al. (2019) to obtain a VAE-based generative model of the barycenter through an iterative procedure. Unfortunately, their fixed point algorithm is not guaranteed to converge. Additionally, the model does not output the OT maps between the input distributions and the barycenter, which therefore, need to be computed in an additional step. Finally, we mention two early generative methods that share some similarities with our approach. Lu et al. (2020) also tackle directly the primal formulation of the OT problem, but enforce the pushforward constraint by adversarial learning (due to their choice of a GAN as generative model) and the bijectivity constraint through a cycle-consistency condition. The model of Xie et al. (2019) also uses a bijective generative model, specifically a neural

ODE model, for the optimal coupling, but they enforce the pushfoward constraint using the Kantorovich-Rubinstein duality for the Wasserstein-1 distance, which results in a minimax optimization problem. Both these models do not address the problem of computing Wasserstein barycenters.

**Advantages of generative models.** Depending on the application of interest, generative models may offer substantial benefits over non-generative ones. For instance, in shape interpolation, the density of the generative model is precisely the interpolating shape and does not require any surface reconstruction from point clouds. In style transfer the generative model can generate previously unseen samples (e.g. new MNIST digits in a given style), as opposed to just transporting already existing ones. In fair regression the generative model effectively performs distributional regression for the fair premium, which allows, for instance, the construction of confidence intervals, the estimation of any statistic of the barycenter distribution, as well as further transformations of the distribution; e.g. the barycenter can be shifted to have mean zero, which is useful, for instance, in actuarial applications.

Since our model is generative, in Section 5.2 we compare with other state-of-the-art generative models, specifically $SC\mathbb{W}_2B$ (Fan et al., 2020) and WIN (Korotin et al., 2019).

## 5. Numerical Experiments

We showcase the performance of our method in a series of numerical experiments. Section 5.1 presents results for the computation of OT maps in high-dimensional settings using Algorithm 1, while Section 5.2 deals with the computation of Wasserstein-2 barycenters on the following data sets: the Swiss roll data set (Section 5.2.1), two high-dimensional location-scatter data sets (Section 5.2.2), the MNIST data set (Section 5.2.3), a high-dimensional Gaussian data set with a large number of input distributions (Section 5.2.4), and a real-life dataset for multivariate fair regression (Section 5.2.5). Hyperparameter choices and all other implementation details are given in Appendix D. Our source code is available online at https://github.com/gvisen/NormalizingFlowsBarycenter.

**Location-scatter families.** Location-scatter families have become standard benchmarks for Wasserstein barycenter models (Korotin et al., 2019; 2021b; Kolesov et al., 2024). They are families of distributions obtained by transforming a base distribution $\mu_0 \in \mathcal{P}(\mathbb{R}^d)$ through invertible affine transformations $f_{M,u} : \mathbb{R}^d \to \mathbb{R}^d$ of the form $f_{M,u} = Mx + u$, for a positive definite matrix $M \in \mathbb{R}^{d \times d}$ and a vector $u \in \mathbb{R}^d$. The Wasserstein-2 barycenter of finitely many distributions belonging to a given location-scatter family can be computed numerically using a fixed-point algorithm (Álvarez-Esteban et al., 2016). In the following experiments,

whenever we mention a location-scatter data set, we will implicitly reproduce the experimental set-up of Korotin et al. (2021b). In particular, we fix $\mathcal{S} = \{1, 2, 3, 4\}$, a vector of weights $w = (w_s)_{s \in \mathcal{S}} = (0.4, 0.3, 0.2, 0.1)$, and choose as base distribution $\mu_0$ either the Swiss roll distribution ($d = 2$), the standard Gaussian distribution $\mathcal{N}(0, I)$ on $\mathbb{R}^d$ or the uniform distribution $\text{Uniform}([-\sqrt{3}, \sqrt{3}]^d)$. We then generate four distributions $(\mu_s)_{s \in \mathcal{S}}$ from the base distribution by setting $\mu_s := f_{M_s, 0 \#} \mu_0$, where $f_{M_s, 0}$ is an affine transformation with positive definite matrix $M_s := R_s^T \Lambda R_s$ and zero shift, $R_s$ is a random rotation matrix sampled uniformly from the Haar measure on $\text{SO}(d)$ and $\Lambda \in \mathbb{R}^{d \times d}$ is a diagonal matrix with diagonal $\left( \frac{1}{2} b^0, \frac{1}{2} b^1, \ldots, \frac{1}{2} b^{d-1} \right)$, for $b = 4^{1/(d-1)}$.

**Metrics.** As proposed in Korotin et al. (2021b), we use the unexplained variance percentage to measure the relative difference between a target distribution $\mu$ and its estimate $\hat{\mu}$:

$$\text{UVP}(\mu, \hat{\mu}) := 100 \cdot \frac{\mathbb{W}_2^2(\mu, \hat{\mu})}{\text{Var}(\mu)} (\%).$$

In a high-dimensional setting it's not possible to compute $\text{UVP}(\mu, \hat{\mu})$ exactly, due to the intractability of $\mathbb{W}_2^2(\mu, \hat{\mu})$. Nevertheless, as shown in Lemma A.2 of Korotin et al. (2019), if $\mu = T_\# \mu_0$ and $\hat{\mu} = \hat{T}_\# \mu_0$, then the following upper bound holds:

$$\text{UVP}(\mu, \hat{\mu}) \leq 100 \frac{\|T - \hat{T}\|_{L^2(\mu_0)}^2}{\text{Var}(T_\# \mu_0)} =: \mathcal{L}^2\text{-UVP}(T, \hat{T}; \mu_0).$$

Additionally, as shown by Dowson & Landau (1982), the following lower bound holds:

$$\text{B}\mathbb{W}_2^2\text{-UVP}(\mu, \hat{\mu}) := 100 \frac{\text{B}\mathbb{W}_2^2(\mu, \hat{\mu})}{\text{Var}(\mu)} \leq \text{UVP}(\mu, \hat{\mu}),$$

where $\text{B}\mathbb{W}_2^2(\mu, \hat{\mu})$ denotes the Bures–Wasserstein metric on Gaussian distributions, computed using the means and covariance matrices of $\mu$ and $\hat{\mu}$.

When solving an OT problem between $\mu_1$ and $\mu_2$ with known OT map $T$, we evaluate the performance of our estimated optimal transport map $\hat{T}$ by computing $\mathcal{L}^2\text{-UVP}(T, \hat{T}; \mu_1)$ and $\text{B}\mathbb{W}_2^2\text{-UVP}(\mu_2, \hat{\mu}_2)$. For barycenter problems, we compute the metrics $\mathcal{L}^2\text{-UVP}(T_s, \hat{T}_s; \mu_s)$ for each $s \in \mathcal{S}$ (where $T_s$ and $\hat{T}_s$ are the true and estimated OT maps from $\mu_s$ to the barycenter $\bar{\mu}$) and then report the mean $\mathcal{L}^2\text{-UVP}$ metric, averaged over all $s \in \mathcal{S}$ using the weights $(w_s)_{s \in \mathcal{S}}$. Additionally, we report the $\text{B}\mathbb{W}_2^2\text{-UVP}(\bar{\mu}, \hat{\bar{\mu}})$ metric, where $\bar{\mu}$ is the true barycenter and $\hat{\bar{\mu}}$ is the model estimate.

### 5.1. Computing OT Maps

We first validate Algorithm 1 by computing OT maps between high-dimensional distributions, for which close-form

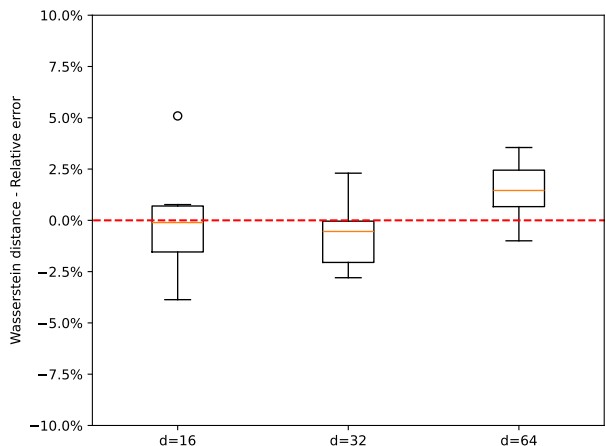

*Figure 1.* $\mathbb{W}_2^2$ relative error on high-dimensional Gaussian data.

solutions are known (see Remark 2.11 in Peyré & Cuturi (2019)). We report the $\mathcal{L}^2\text{-UVP}$ and $\text{B}\mathbb{W}_2^2\text{-UVP}$ metrics in Table 1 and Table 2, for Gaussian and uniform distributions respectively, together with their standard deviation from ten trials. In terms of Wasserstein-2 distance, in Figure 1 we report a box plot of the (signed) relative error of the estimated distance computed on high-dimensional Gaussian data for ten random initializations of our model. All results show a very good performance in high-dimensions.

### 5.2. Computing Wasserstein-2 Barycenters

#### 5.2.1. SWISS ROLL DATASET

We begin by evaluating Algorithm 2 qualitatively on a two-dimensional location-scatter data set with a Swiss roll base distribution, $\mathcal{S} = \{1, 2, 3, 4\}$ and weights $w = (0.4, 0.3, 0.2, 0.1)$. Figure 2 shows samples from the true input distributions $(\mu_s)_{s \in \mathcal{S}}$ (first row) and samples generated from our trained model (second row). At convergence, all input distributions have been learned well, which implies that the pushforward constraints are fully satisfied. In Figure 3 we compare samples from the true barycenter and the learned barycenter, i.e. $h_\# \lambda$. Unlike other generative models (see Figure 1 in Korotin et al. (2021b) for a comparison with $\text{SC}\mathbb{W}_2\text{B}$), our learned barycenter distribution is sharp and reproduces the highly non-linear structure of the Swiss roll distribution. We also verify in Figure 4 that the learned OT maps correctly transport samples from the input distributions to the barycenter via the transformations $x \mapsto h(f_\theta^{-1}(x, s))$, for all $s \in \mathcal{S}$.

#### 5.2.2. HIGH-DIMENSIONAL EXPERIMENTS

We now compare our method's performance to other state-of-the-art models on two location-scatter benchmark

*Table 1.* $\mathcal{L}^2$-UVP and $\mathbb{BW}_2^2$-UVP metrics for our method when computing OT maps between high-dimensional Gaussian distributions.

| METRIC | $d = 2$ | $d = 4$ | $d = 8$ | $d = 16$ | $d = 32$ | $d = 64$ | $d = 128$ |
|---|---|---|---|---|---|---|---|
| $\mathcal{L}^2$-UVP | $0.004 \pm 0.001$ | $0.086 \pm 0.001$ | $0.204 \pm 0.002$ | $0.623 \pm 0.004$ | $2.389 \pm 0.018$ | $1.658 \pm 0.005$ | $3.102 \pm 0.009$ |
| $\mathbb{BW}_2^2$-UVP | $0.002 \pm 0.001$ | $0.015 \pm 0.005$ | $0.008 \pm 0.001$ | $0.014 \pm 0.001$ | $0.034 \pm 0.002$ | $0.054 \pm 0.002$ | $0.088 \pm 0.002$ |

*Table 2.* $\mathcal{L}^2$-UVP and $\mathbb{BW}_2^2$-UVP metrics for our method when computing OT maps between high-dimensional uniform distributions.

| METRIC | $d = 2$ | $d = 4$ | $d = 8$ | $d = 16$ | $d = 32$ | $d = 64$ | $d = 128$ |
|---|---|---|---|---|---|---|---|
| $\mathcal{L}^2$-UVP | $0.54 \pm 0.006$ | $1.506 \pm 0.019$ | $2.046 \pm 0.031$ | $1.404 \pm 0.006$ | $4.255 \pm 0.016$ | $3.833 \pm 0.014$ | $7.137 \pm 0.012$ |
| $\mathbb{BW}_2^2$-UVP | $0.007 \pm 0.001$ | $0.029 \pm 0.003$ | $0.047 \pm 0.004$ | $0.036 \pm 0.003$ | $0.704 \pm 0.009$ | $0.738 \pm 0.01$ | $1.223 \pm 0.005$ |

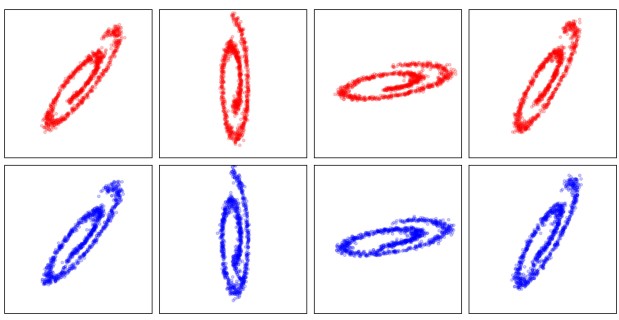

*Figure 2.* Samples from the true input distributions (first row) and from the input distributions as learned by our model (second row).

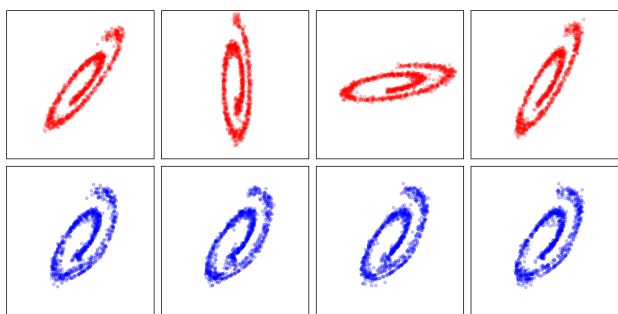

*Figure 4.* Samples from the input distributions $\mu_s$ (first row) and their image after being transported to the barycenter (second row).

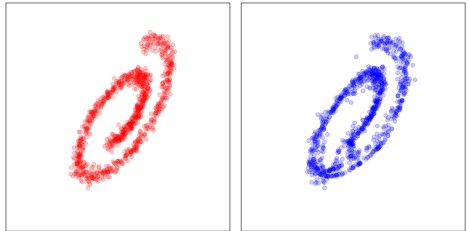

*Figure 3.* True barycenter (left) and learned barycenter (right).

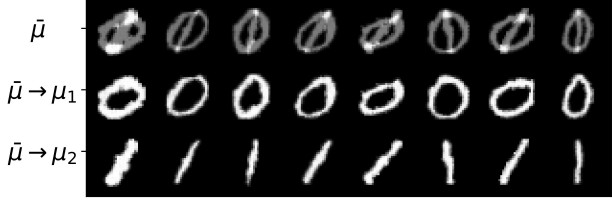

*Figure 5.* Sample from learned barycenter transported to the two input distributions.

datasets across a range of input dimensions (from $d = 2$ to $d = 128$): a Gaussian data set (see Table 3) and a uniform one (see Table 4). Our model has better performance than the competitors, SC$\mathbb{W}_2$B (Fan et al., 2020) and WIN (Korotin et al., 2022), with better or equal training time (for more details, see Appendix D).

### 5.2.3. MNIST DIGITS

Next, we test our model on image data using the MNIST data set, using our conditional Glow architecture. We first compute the barycenter of the empirical distributions of the digits zero and one. We recover the well-known result that

each barycenter sample is the average of images from the input measures (Korotin et al., 2019). In Figure 5 we show samples from the learned barycenter and their images when transported to the two input distributions. While Figure 6 and Figure 7 show the inverse operation on samples from the true input distributions.

We also train our model using as input distributions the empirical distributions of all ten digits. Thanks to our model's conditional architecture, the training time scales well in the number of input distributions, therefore solving this Wasserstein barycenter task did not take more training time than the previous one. In Figure 8 we show samples from the

*Table 3.* $\mathcal{L}^2$-UVP and $\mathbb{BW}_2^2$-UVP metrics on location-scatter Gaussian data (from Table 5 in Korotin et al. (2019) for SC$\mathbb{W}_2$B and WIN).

| METRIC | METHOD | $d = 2$ | $d = 4$ | $d = 8$ | $d = 16$ | $d = 32$ | $d = 64$ | $d = 128$ |
|---|---|---|---|---|---|---|---|---|
| $\mathcal{L}^2$-UVP | OURS | $0.025 \pm 0.0$ | $0.079 \pm 0.002$ | $0.076 \pm 0.001$ | $0.1 \pm 0.001$ | $0.382 \pm 0.001$ | $0.544 \pm 0.001$ | $1.55 \pm 0.002$ |
| $\mathbb{BW}_2^2$-UVP | SC$\mathbb{W}_2$B | 0.070 | 0.090 | 0.160 | 0.280 | 0.430 | 0.590 | 1.280 |
| | WIN | 0.010 | 0.020 | 0.010 | 0.080 | 0.110 | 0.230 | 0.380 |
| | OURS | $0.005 \pm 0.001$ | $0.004 \pm 0.001$ | $0.004 \pm 0.001$ | $0.007 \pm 0.001$ | $0.024 \pm 0.001$ | $0.046 \pm 0.001$ | $0.094 \pm 0.001$ |

*Table 4.* $\mathcal{L}^2$-UVP and $\mathbb{BW}_2^2$-UVP metrics on location-scatter uniform data (from Table 5 in Korotin et al. (2019) for SC$\mathbb{W}_2$B and WIN).

| METRIC | METHOD | $d = 2$ | $d = 4$ | $d = 8$ | $d = 16$ | $d = 32$ | $d = 64$ | $d = 128$ |
|---|---|---|---|---|---|---|---|---|
| $\mathcal{L}^2$-UVP | OURS | $0.998 \pm 0.016$ | $0.399 \pm 0.005$ | $0.498 \pm 0.003$ | $0.697 \pm 0.002$ | $1.301 \pm 0.002$ | $2.428 \pm 0.002$ | $5.506 \pm 0.004$ |
| $\mathbb{BW}_2^2$-UVP | SC$\mathbb{W}_2$B | 0.120 | 0.100 | 0.190 | 0.290 | 0.460 | 0.600 | 1.380 |
| | WIN | 0.040 | 0.060 | 0.060 | 0.080 | 0.110 | 0.270 | 0.460 |
| | OURS | $0.01 \pm 0.004$ | $0.053 \pm 0.007$ | $0.033 \pm 0.005$ | $0.058 \pm 0.003$ | $0.081 \pm 0.002$ | $0.159 \pm 0.002$ | $0.362 \pm 0.003$ |

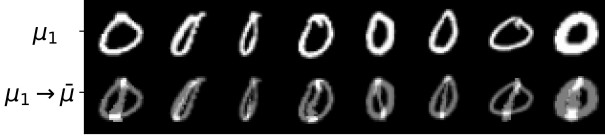

*Figure 6.* Sample from the first input distribution transported to the barycenter.

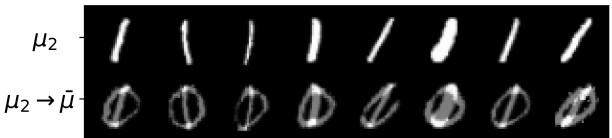

*Figure 7.* Sample from the second input distribution transported to the barycenter.

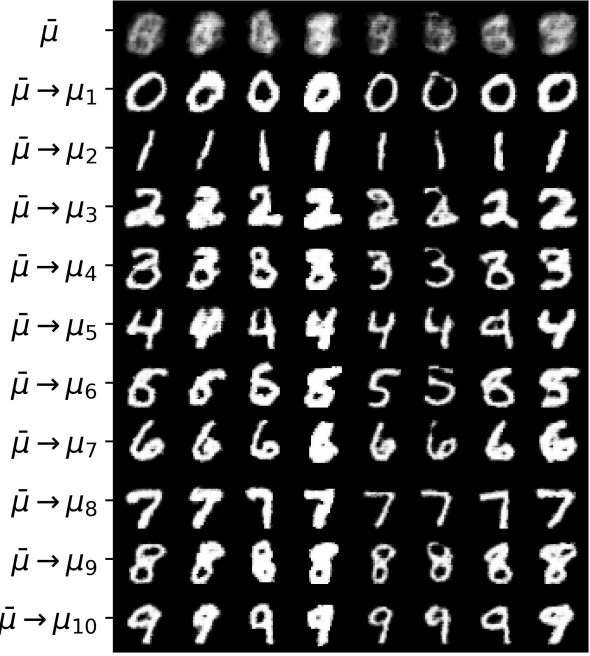

*Figure 8.* Sample from learned barycenter transported to all input distributions.

learned barycenter and their images when transported to all ten input distributions.

We illustrate a style translation application of our model in Figure 9, where we transport samples from a source distribution (e.g. the distribution of zeros), through the latent space, to a target distribution (e.g. the distribution of ones). Under this transformation, stylistic features of the zero digit – such as its slant and thickness – are preserved, showing that the latent space in our model encodes structural features of the input data and can be used to transfer them.

### 5.2.4. LARGE NUMBER OF INPUT DISTRIBUTIONS

In this section, we leverage the conditional architecture of our model to compute Wasserstein-2 barycenters for a large number of input distributions. Specifically, we evaluate

our model on high-dimensional Gaussian data sets ($d = 64$) with various numbers of input distributions ($n = 8, 64, 128$). Different input distributions are indexed by $\mathcal{S}$, the uniform grid on $[0, \pi]$ with $n$ points, and are zero-mean Gaussian distributions with covariance $\Sigma_s = R(s)^T \Sigma_0 R(s)$, where $\Sigma_0$ is a diagonal matrix with diagonal $(2, 1/2, \ldots, 1/2)$ and $R(s)$ is a rotation matrix describing a rotation in the first two coordinates by an angle $s \in \mathcal{S} \subset [0, \pi]$. In this way we can generate a large number of input distributions that vary smoothly in the univariate parameter $s$. We then com-

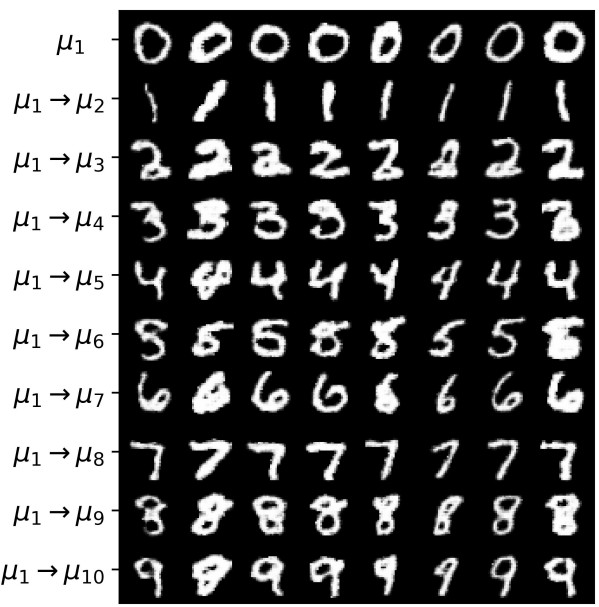

*Figure 9.* Sample from first input distribution transported to all other input distributions.

pute the Wasserstein-2 barycenters of these distributions using Algorithm 2. The results are presented in Table 5 and show no degradation in performance as the number of input distributions increases.

*Table 5.* Performance as number of input distributions increases.

| METRIC | $n = 4$ | $n = 16$ | $n = 64$ | $n = 128$ |
|---|---|---|---|---|
| $\mathcal{L}^2$-UVP | 0.057 | 0.119 | 0.081 | 0.073 |
| $B\mathbb{W}_2^2$-UVP | 0.053 | 0.031 | 0.016 | 0.051 |

### 5.2.5. MULTIVARIATE FAIR REGRESSION

Fair regression, in the sense of demographic parity, looks for a regression function $f(X, S)$ that minimizes the cost $\mathbb{E}[\|Y - f(X, S)\|^2]$, such that $f(X, S)$ is independent of $S$. In this context $S$ denotes sensitive features, such as race or gender, with respect to which discrimination is not allowed. The solution to this constrained optimization is precisely the Wasserstein-2 barycenter of the conditional distributions of $Y$ given $S$ (Chzhen et al., 2020; Gouic et al., 2020). To demonstrate the applicability of our model to multivariate fair regression, we work on the dataset "Communities and Crime" (Redmond, 2002), which is a benchmark dataset in the fairness literature, and we regress the target variables "percentage of officers assigned to drug units" and "total number of violent crimes per 100k population" (target variable $Y$) on 127 socio-economic features (non-sensitive

features $X$) and the percentage of the population that is African-American (sensitive feature $S$ with range $[0, 1]$).

As shown in Table 6, a standard regression leads to strong correlation between the first predicted variable and the sensitive feature for several correlation measures, while our fair regression achieves almost perfect uncorrelatedness. In all cases, we fail to reject the null hypothesis of no association.

*Table 6.* Correlation between regressor and sensitive feature.

| CORRELATION COEFFICIENT | STANDARD REGRESSION | FAIR REGRESSION |
|---|---|---|
| PEARSON | 0.43 | 0.0003 (P-VALUE: 0.99) |
| SPEARMAN | 0.47 | 0.0058 (P-VALUE: 0.80) |
| KENDALL | 0.32 | 0.0039 (P-VALUE: 0.80) |

We point out that this experiment requires computing the barycenter of 100 input distributions (the cardinality of the range of $S$, rounded up to the nearest percentage point), which would be computationally challenging for any other numerical Wasserstein barycenter method.

## 6. Conclusion

We have introduced a new method for computing OT maps in high-dimensional settings using conditional normalizing flows. Unlike previous methods, we avoid complex adversarial optimization and directly solve the primal OT problem using gradient-based minimization of the transport cost. We have shown how the approach can be extended to compute Wasserstein barycenters by solving a conditional variance minimization problem, and we have compared to state-of-the-art competitor models on several benchmarks. Our approach yields accurate results and can be used to compute Wasserstein barycenters for hundreds of input distributions, which was computationally infeasible with previous methods.

## 7. Limitations

A limitation of our approach is that Algorithm 2 applies only to Wasserstein-2 barycenters. Its extension to Wasserstein-$p$ barycenters for $p \neq 2$ is currently still an open research question. Furthermore, training might be computationally intensive on image datasets much larger than the ones we tested on, i.e. for $d \gg 1000$.

## Impact Statement

This paper presents work whose goal is to advance the field of Machine Learning. There are many potential societal consequences of our work, none which we feel must be

specifically highlighted here.

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

## A. Auxiliary Results

The following are some standard results from measure and integration theory that are used in this paper.

**Lemma A.1** (Change of variables). *Let $f : X \to Y$ be a measurable function between two measurable spaces $(X, \mathcal{A})$ and $(Y, \mathcal{B})$. If $\mu$ is probability measure on $(X, \mathcal{A})$, then a measurable function $g : Y \to \mathbb{R}$ satisfies $g \in L^1(f_{\#}\mu)$ if and only if $g \circ f \in L^1(\mu)$. Moreover, in this case one has*

$$\int_Y g(y)(f_{\#}\mu)(dy) = \int_X g(f(x))\mu(dx).$$

*Proof.* See Theorem 3.6.1 in Bogachev & Ruas (2007). □

**Lemma A.2.** *Let $f : \mathbb{R}^d \to \mathbb{R}^d$ be a Borel function and $\mu$ a Borel probability measure on $\mathbb{R}^d$. Then $f = 0$ $\mu$-almost surely if and only if $\int_{\mathbb{R}^d} \|f(x)\|\mu(dx) = 0$.*

*Proof.* $\int_{\mathbb{R}^d} \|f(x)\|\mu(dx) = 0$ if and only if $\|f(x)\| = 0$ $\mu$-almost surely (see, for instance, Proposition 2.16 in Folland (1999)), which is the case if and only if $f(x)$ is the zero vector $\mu$-almost surely. □

**Lemma A.3.** *Let $\mu, \nu \in \mathcal{P}_p(\mathbb{R}^d)$ for some $p \geq 1$. If $f \in B(\mu, \nu)$ and $\tilde{f}$ is a $(\mu, \nu)$-inverse of $f$, then*

(i) $\|f\| \in L^p(\mu)$,

(ii) $\tilde{f}$ is a $\nu$-a.s. unique $(\mu, \nu)$-inverse of $f$,

(iii) $\tilde{f} \in B(\nu, \mu)$, $\|\tilde{f}\| \in L^p(\nu)$ and $f$ is a $\mu$-a.s. unique $(\nu, \mu)$-inverse of $\tilde{f}$.

*Proof.* (i) Let $f \in B(\mu, \nu)$. Then by Lemma A.1,

$$\int_{\mathbb{R}^d} \|f(x)\|^p\mu(dx) = \int_{\mathbb{R}^d} \|y\|^p\nu(dy) < \infty.$$

(ii) Assume $\bar{f}$ is another $(\mu, \nu)$-inverse of $f$, then

$$
\begin{aligned}
0 &= \int_{\mathbb{R}^d} \|(\tilde{f} \circ f)(x) - x\|\, \mu(dx) && (\tilde{f} \circ f = \text{id}, \mu\text{-a.s.}) \\
&= \int_{\mathbb{R}^d} \|(\tilde{f} \circ f)(x) - (\bar{f} \circ f)(x)\|\, \mu(dx) && (\bar{f} \circ f = \text{id}, \mu\text{-a.s.}) \\
&= \int_{\mathbb{R}^d} \|\tilde{f}(y) - \bar{f}(y)\|\, \nu(dy), && (\text{change of variable } y = f(x), \text{ see Lemma A.1})
\end{aligned}
$$

which, by Lemma A.2, implies that $\tilde{f} = \bar{f}$ $\nu$-a.s.

(iii) We first show that $\tilde{f}_{\#}\nu = \mu$. Let $A \in \mathcal{B}(\mathbb{R}^d)$, then

$$
\begin{aligned}
\nu(\tilde{f}^{-1}(A)) &= \mu(f^{-1}(\tilde{f}^{-1}(A))) && (f_{\#}\mu = \nu) \\
&= \mu((\tilde{f} \circ f)^{-1}(A)) \\
&= \mu(A). && (\tilde{f} \circ f = \text{id}, \mu\text{-a.s.})
\end{aligned}
$$

Since $\tilde{f}$ is the $(\mu, \nu)$-inverse of $f$, clearly $f$ satisfies the definition of a $(\nu, \mu)$-inverse of $\tilde{f}$, therefore $\tilde{f}$ belongs to $B(\nu, \mu)$. Finally, $\|\tilde{f}\| \in L^p(\nu)$ simply follows from $\tilde{f} \in B(\nu, \mu)$ and (i). □

**Lemma A.4.** *Let $\lambda, \mu, \nu \in \mathcal{P}_p(\mathbb{R}^d)$. If $f \in B(\lambda, \mu)$ and $g \in B(\mu, \nu)$, then $g \circ f \in B(\lambda, \nu)$. Furthermore, if $\tilde{f}$ is a $(\lambda, \mu)$-inverse of $f$ and $\tilde{g}$ is a $(\mu, \nu)$-inverse of $g$, then $\tilde{f} \circ \tilde{g}$ is a $(\lambda, \nu)$-inverse of $g \circ f$.*

*Proof.* Clearly $(g \circ f)_{\#}\lambda = g_{\#}(f_{\#}\lambda) = g_{\#}\mu = \nu$. Furthermore,

$$
\begin{aligned}
0 &= \int_{\mathbb{R}^d} \|(\tilde{f} \circ \tilde{g})(y) - (\tilde{f} \circ \tilde{g})(y)\| \, \nu(dy) \\
&= \int_{\mathbb{R}^d} \|(\tilde{f} \circ \tilde{g} \circ g \circ \tilde{g})(y) - (\tilde{f} \circ \tilde{g})(y)\| \, \nu(dy) && (g \circ \tilde{g} = \text{id } \nu\text{-a.s.}) \\
&= \int_{\mathbb{R}^d} \|(\tilde{f} \circ \tilde{g} \circ g)(x) - \tilde{f}(x)\| \, \mu(dx) && (\text{change of variables } x = \tilde{g}(y)) \\
&= \int_{\mathbb{R}^d} \|(\tilde{f} \circ \tilde{g} \circ g \circ f \circ \tilde{f})(x) - \tilde{f}(x)\| \, \mu(dx) && (f \circ \tilde{f} = \text{id } \mu\text{-a.s.}) \\
&= \int_{\mathbb{R}^d} \|(\tilde{f} \circ \tilde{g} \circ g \circ f)(z) - z\| \, \lambda(dz), && (\text{change of variables } z = \tilde{f}(x))
\end{aligned}
$$

which shows that $\tilde{f} \circ \tilde{g} \circ g \circ f = \text{id } \lambda$-almost surely by Lemma A.2. An analogous computation shows that $g \circ f \circ \tilde{f} \circ \tilde{g} = \text{id }$ $\nu$-almost surely. $\qquad\square$

## B. Proofs

**Proof of Lemma 3.1.** Existence of a unique solution $\gamma \in \Gamma(\mu, \nu)$ to Kantorovich's problem (2) and the fact that $\gamma$ is of the form $(\text{id}, T)_{\#}\mu$ for a measurable map $T : \mathbb{R}^d \to \mathbb{R}^d$ pushing $\mu$ to $\nu$, follows from Theorem 3.7 of Gangbo & McCann (1996). That $T$ is $\mu$-almost surely unique and belongs to $B(\mu, \nu)$ is a consequence of Theorem 4.5 of Gangbo & McCann (1996). $\qquad\square$

The proofs of Theorem 3.2 and Theorem 3.3 rely on the following lemma.

**Lemma B.1.** *Consider probability measures $\lambda, \mu, \nu \in \mathcal{P}_{p,ac}(\mathbb{R}^d)$ for some $p > 1$. Then $B(\lambda, \mu)$ is non-empty, and for every $f \in B(\lambda, \mu)$, there exists a $g \in B(\lambda, \nu)$ such that*

$$
\mathbb{W}_p^p(\mu, \nu) = \int_{\mathbb{R}^d} \|f(z) - g(z)\|^p \lambda(dz) = \int_{\mathbb{R}^d} \|x - g \circ \tilde{f}(x)\|^p \mu(dz), \tag{5}
$$

*where $\tilde{f}$ is any $(\lambda, \mu)$-inverse of $f$. In particular, $g \circ \tilde{f}$ is an optimal transport map between $\mu$ and $\nu$.*

*Proof.* It follows from Theorems 3.7 and 4.5 of Gangbo & McCann (1996) that $B(\lambda, \mu)$ is non-empty and there exists a $T \in B(\mu, \nu)$ such that

$$
\int_{\mathbb{R}^d} \|x - T(x)\|^p \, \mu(dx) = \inf_{\pi \in \Pi(\mu, \nu)} \int_{\mathbb{R}^d \times \mathbb{R}^d} \|x - y\|^p \, \pi(dx, dy) = \mathbb{W}_p^p(\mu, \nu).
$$

So, for a given $f \in B(\lambda, \mu)$, the mapping $g = T \circ f$ is in $B(\lambda, \nu)$ by Lemma A.4 and, by the change-of-variable-formula,

$$
\int_{\mathbb{R}^d} \|f(z) - g(z)\|^p \lambda(dz) = \int_{\mathbb{R}^d} \|f(z) - T \circ f(z)\|^p \lambda(dz) = \int_{\mathbb{R}^d} \|x - T(x)\|^p \mu(dx) = \mathbb{W}_p^p(\mu, \nu).
$$

Since $T = g \circ \tilde{f}$ $\mu$-almost surely, this shows (5), and $g \circ \tilde{f}$ is an optimal transport map between $\mu$ and $\nu$, which completes the proof of the lemma. $\qquad\square$

**Proof of Theorem 3.2.** We know from Lemma 3.1 that there exists an optimal transport map $T \in B(\mu, \nu)$. So, for any $f \in B(\lambda, \mu)$ and $g \in B(\lambda, \nu)$, we have

$$
\|f - g\|_{L^p(\lambda)}^p = \int_{\mathbb{R}^d} \|x - g \circ \tilde{f}(x)\|^p \mu(dx) \geq \int_{\mathbb{R}^d} \|x - T(x)\|^p \mu(dx) = \mathbb{W}_p^p(\mu, \nu), \tag{6}
$$

where $\tilde{f}$ is any $(\lambda, \mu)$-inverse of $f$ and we used the fact that $g \circ \tilde{f}$ pushes $\mu$ to $\nu$ by Lemma A.4.

On the other hand, it follows from Lemma B.1 that there exist $f \in B(\lambda, \mu)$ and $g \in B(\lambda, \nu)$ such that the inequality in (6) becomes an equality, which completes the proof. □

**Proof of Theorem 3.3.** It follows from Brizzi et al. (2025) that the $w$-weighted Wasserstein-2 barycenter $\bar{\mu}$ of $(\mu_s)_{s \in \mathcal{S}}$ belongs to $\mathcal{P}_{2,ac}(\mathbb{R}^d)$. Moreover, we obtain from the definitions of the barycenter $\bar{\mu}$ and the Wasserstein-2 distance that

$$\sum_{s \in \mathcal{S}} w_s \mathbb{W}_2^2(\bar{\mu}, \mu_s) \leq \sum_{s \in \mathcal{S}} w_s \mathbb{W}_2^2(h_\# \lambda, \mu_s) \leq \sum_{s \in \mathcal{S}} w_s \int_{\mathbb{R}^d} \|h(z) - f(z,s)\|^2 \lambda(dz)$$

for all Borel maps $h : \mathbb{R}^d \to \mathbb{R}^d$ and $f : \mathbb{R}^d \times \mathcal{S} \to \mathbb{R}^d$ such that $f(\cdot, s)$ transports $\lambda$ to $\mu_s$ for every $s \in \mathcal{S}$. On the other hand, we know from Proposition B.1 that there exist $h \in B(\lambda, \bar{\mu})$ and $f(\cdot, s) \in B(\lambda, \mu_s)$, $s \in \mathcal{S}$, such that

$$\mathbb{W}_2^2(\bar{\mu}, \mu_s) = \int_{\mathbb{R}^d} \|h(z) - f(z,s)\|^2 \lambda(dz) \quad \text{for all } s \in \mathcal{S},$$

and therefore,

$$\sum_{s \in \mathcal{S}} w_s \mathbb{W}_2^2(\bar{\mu}, \mu_s) = \sum_{s \in \mathcal{S}} w_s \mathbb{W}_2^2(h_\# \lambda, \mu_s) = \sum_{s \in \mathcal{S}} w_s \int_{\mathbb{R}^d} \|h(z) - f(z,s)\|^2 \lambda(dz).$$

In particular, $h$ minimizes

$$\sum_{s \in \mathcal{S}} w_s \int_{\mathbb{R}^d} \|h(z) - f(z,s)\|^2 \lambda(dz) = \mathbb{E}\left[\|h(Z) - f(Z,S)\|^2\right]$$

over all Borel maps from $\mathbb{R}^d$ to $\mathbb{R}^d$, which implies that it is given by the component-wise conditional expectation

$$h(Z) = \mathbb{E}\left[f(Z,S) \mid Z\right] = \sum_{s \in \mathcal{S}} w_s f(Z,s);$$

see, e.g. Durrett (2019). Moreover, $f$ minimizes

$$\sum_{i=1}^d \mathbb{E}\left[(\mathbb{E}\left[f_i(Z,S) \mid Z\right] - f_i(Z,S))^2\right] = \sum_{i=1}^d \mathbb{E}\left[\text{Var}(f_i(Z,S) \mid Z)\right]$$

over all mappings $f : \mathbb{R}^d \times \mathcal{S} \to \mathbb{R}^d$ satisfying $f(., s) \in B(\lambda, \mu_s)$ for all $s \in \mathcal{S}$. This proves (i).

Now, assume $\tilde{f} : \mathbb{R}^d \times \mathcal{S} \to \mathbb{R}^d$ is another function such that $\tilde{f}(\cdot, s) \in B(\lambda, \mu_s)$ for all $s \in \mathcal{S}$ and

$$\sum_{i=1}^d \mathbb{E}\left[\text{Var}(\tilde{f}_i(Z,S) \mid Z)\right] = \sum_{i=1}^d \mathbb{E}\left[\text{Var}(f_i(Z,S) \mid Z)\right].$$

Then, for $\tilde{h}(Z) = \mathbb{E}[\tilde{f}(Z,S) \mid Z] = \sum_{s \in \mathcal{S}} w_s \tilde{f}(Z,s)$, one has

$$\sum_{s \in \mathcal{S}} w_s \mathbb{W}_2^2(\tilde{h}_\# \lambda, \mu_s) \leq \sum_{s \in \mathcal{S}} w_s \int_{\mathbb{R}^d} \|\tilde{h}(z) - \tilde{f}(z,s)\|^2 \lambda(dz) = \sum_{i=1}^d \mathbb{E}\left[\text{Var}(\tilde{f}_i(Z,S) \mid Z)\right]$$

$$= \sum_{i=1}^d \mathbb{E}\left[\text{Var}(f_i(Z,S) \mid Z)\right] = \sum_{s \in \mathcal{S}} w_s \int_{\mathbb{R}^d} \|h(z) - f(z,s)\|^2 \lambda(dz) = \sum_{s \in \mathcal{S}} w_s \mathbb{W}_2^2(\bar{\mu}, \mu_s).$$

But since $\bar{\mu}$ is the unique $w$-weighted Wasserstein-2 barycenter, we conclude that $\tilde{h}_\# \lambda = \bar{\mu}$, which shows (ii). □

## C. Background on Normalizing Flows

Our method relies on conditional normalizing flows to compute OT maps and Wasserstein barycenters. For the reader's convenience, we collect here all the necessary background on normalizing flows. For more information, we point the reader to Kobyzev et al. (2020) and Papamakarios et al. (2021), which are two recent, well-written survey papers.

Normalizing flows are generative models used to learn high-dimensional data distributions by transforming a simple latent distribution (e.g. a standard normal distribution) through a sequence of diffeomorphisms. A key advantage of normalizing flows is that, unlike other generative models such as VAEs and GANs, density evaluations of the model distribution are efficient, thanks to the change of variables formula, which allows efficient learning by likelihood maximization. More precisely, if $Z$ is an $\mathbb{R}^d$-valued random variable with density $p_Z$ and $f : \mathbb{R}^d \to \mathbb{R}^d$ is a composition of diffeomorphisms $f = f_1 \circ \ldots \circ f_N$, then the random variable $X = f(Z)$ has density:

$$p(x) = p_Z(f^{-1}(x)) \, |\det(D_x f^{-1}(x))|, \tag{7}$$

where $f^{-1} = f_N^{-1} \circ \ldots \circ f_1^{-1}$ and

$$|\det(D_x f^{-1}(x))| = \prod_{i=1}^{N} |\det(D_x f_i^{-1}(x))|.$$

The challenge in designing good normalizing flows consists in finding flows that have a tractable Jacobian and are easy to invert, but are expressive enough to be able to approximate interesting data distributions.

**Real NVP.** Real NVP was the first normalizing flow model with competitive performance on benchmark image data sets (Dinh et al., 2016). It leverages coupling layers, first introduced by Dinh et al. (2014), to provide an economical way of representing highly expressive transformations. In a coupling layer the input is first split into two components $z = (z^A, z^B)$, then $z^A$ is transformed using a simple parametric bijective transformation, whose parameters depend (possibly in a highly non-linear way) on $z^B$, as follows:

$$\begin{cases} x^A & = f_{\theta(z^B)}(z^A) \\ x^B & = z^B \end{cases}$$

where $f_\theta(\cdot)$ is a parametric bijection with parameters $\theta$, called a *coupling function*. In particular, Real NVP uses affine coupling layers with log-scale and shift parameters, $\theta(z^B) = (a(z^B), b(z^B))$, resulting in a coupling function of the form:

$$f_{\theta(z^B)}(z^A) = z^A \odot \exp(a(z^B)) + b(z^B),$$

where $a(\cdot)$ and $b(\cdot)$ are feedforward or convolutional neural networks.

Stacking many affine coupling layers and permutation layers (i.e. bijective transformations $z \mapsto \pi(z)$, for a random permutation $\pi$ fixed at initialization) results in highly complex, non-linear transformations.

**Glow.** Kingma & Dhariwal (2018) proposed a normalizing flow model with improved performance on image data. They replaced the random permutation layers of Real NVP with invertible 1x1 convolutions, thereby allowing the model to learn the permutations, instead of randomly fixing them at initialization. They also added activation normalization layers, which adapt the batch normalization technique to the small mini-batch regime typical of training on high-resolution image data.

**Multi-scale architecture.** A common problem of normalizing flows is that the dimension of the latent space must match the dimension of the target distribution to be learned, due to the bijectivity constraint. In high-dimensional settings, this results in very deep and large models, that may be difficult to train. Multi-scale architectures solve this problem by exploiting the fact that empirical data distributions, especially for image data, tend to concentrate around low dimensional manifolds (Gong et al., 2019; Pope et al., 2021; Brown et al., 2022; Loaiza-Ganem et al., 2024). This is done by dividing the model's flows into $L$ separate *scales* or *levels*, as shown in Figure 10, consisting of $K$ layers each, with latent dimensions being gradually added at each subsequent scale. In this way, only a low-dimensional latent vector $z$ is pushed through the full depth of the model, while the remaining dimensions undergo simpler, easier-to-train transformations, capturing finer details.

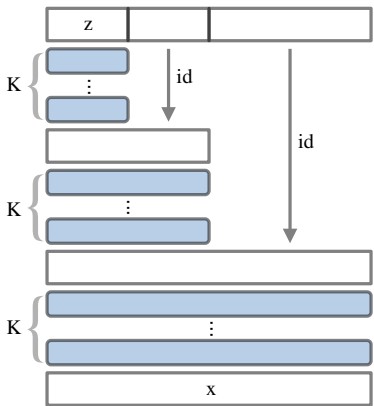

*Figure 10.* Multi-scale architecture with $L = 3$ scales of $K$ flows each.

## D. Experimental details

Table 7 presents all hyperparameter choices for our model in all numerical experiments. The architecture type (either conditional Real NVP or conditional Glow) reports the number of hidden neurons in each hidden layer of the conditioner network. $L$ and $K$ denote the number of scales and flows per scale in the multi-scale architecture.

*Table 7.* Hyperparameter choices for our model for all numerical experiments.

| DATASET | OPT | LR | ITERS | BS | TYPE | | $L$ | $K$ | | WEIGHTS |
|---|---|---|---|---|---|---|---|---|---|---|
| SWISS ROLL | ADAM | $10^{-4}$ | $5 \cdot 10^3$ | $10^4$ | REAL [64, 64] | NVP | 1 | 32 | | LOGSPACE $[0, -4]$ |
| HIGH-DIM GAUSSIAN ($d = 2 - 128$) | ADAM | $10^{-3}$ | $10^4$ | $10^4$ | REAL [64, 64] | NVP | $\log_2(d)$ | 32 FOR $d = 2$ 16 FOR $d = 4, 8, 16$ 8 FOR $d \geq 32$ | | LOGSPACE $[0, -2]$ |
| HIGH-DIM UNIFORM ($d = 2 - 128$) | ADAM | $10^{-4}$ | $10^4$ | $10^4$ | REAL [64, 64] | NVP | $\log_2(d)$ | 32 FOR $d = 2$ 16 FOR $d = 4, 8, 16$ 8 FOR $d \geq 32$ | | LOGSPACE $[0, -2]$ |
| MNIST | ADAM | $10^{-4}$ | $5 \cdot 10^4$ | 32 | GLOW, CHANNELS | 256 | 4 | 16 | | LOGSPACE $[1, -2]$ |
| LARGE NUMBER OF INPUT DISTRIBUTIONS | ADAM | $10^{-3}$ | $10^4$ | $10^3$ | REAL [64, 64] | NVP | $\log_2(d)$ | 32 | | LOGSPACE $[0, -2]$ |

All numerical experiments were run on an NVIDIA GeForce RTX 4090 GPU with 24 GB of memory, except for the MNIST data set experiment, which was run on an NVIDIA RTX 6000 Ada with 48 GB of memory. Our implementation is written in Python, is both GPU and CPU-compatible, and builds on `PyTorch`, the `normflows` package by Stimper et al. (2023) and the code repository by Korotin et al. (2021b).

The training times for the location-scatter experiments in Section 5.2.2 are shown in Table 8 and Table 9. We emphasize, though, that in practice all methods converge in approximately 10-20 minutes, with longer times needed only for top-notch performance. On the MNIST data set our method converges in approximately 30 minutes.

*Table 8.* Training times on high-dimensional location-scatter Gaussian data.

| METHOD | $d = 2$ | $d = 4$ | $d = 8$ | $d = 16$ | $d = 32$ | $d = 64$ | $d = 128$ |
|---|---|---|---|---|---|---|---|
| SC$\mathbb{W}_2$B | 114M 9S | 110M51S | 108M21S | 108M 4S | 108M52S | 109M36S | 114M56S |
| WIN | 60M 9S | 57M56S | 58M32S | 75M53S | 57M26S | 92M54S | 93M28S |
| OURS | 23M32S | 23M58S | 35M28S | 47M 7S | 30M 1S | 35M59S | 42M 3S |

*Table 9.* Training times on high-dimensional location-scatter Uniform data.

| METHOD | $d = 2$ | $d = 4$ | $d = 8$ | $d = 16$ | $d = 32$ | $d = 64$ | $d = 128$ |
|---|---|---|---|---|---|---|---|
| SC$\mathbb{W}_2$B | 114M 9S | 110M51S | 108M21S | 108M 4S | 108M52S | 109M36S | 114M56S |
| WIN | 90M17S | 58M55S | 59M45S | 60M45S | 60M 8S | 93M16S | 103M13S |
| OURS | 110M 1S | 114M36S | 110M46S | 111M14S | 110M11S | 108M55S | 111M49S |

