# OpenReview forum: "Computing Optimal Transport Maps and Wasserstein Barycenters Using Conditional Normalizing Flows"
_ICML.cc/2025/Conference — ICML 2025 poster_

### Official Review · Reviewer_aK7B · 2025-03-10

**Overall Recommendation:** 3

**Summary:**

An alternative formulation of the p-Wasserstein distance is introduced and it consists in a constrained L^p minimisation problem for a given latent distribution. This alternative formulation allows the authors to both i) directly solve the Wasserstein primal minimization problem via stochastic gradient decent and ii) exploit conditional normalizing flows in order to compute the optimal transport map (already exiting here, since the source and destination measures are absolutely continuous w.r.t. to the Lebesgue measure) as well as the Wasserstein barycenters. Two pseudo-algorithms are well detailed and experiments on two datasets (Swiss roll and Mnist) are used to support the two main claims: the proposed approach allows one to compute barycenters from hundreds of input distributions and it is efficient in high-dimensional spaces.

**Claims And Evidence:**

From a theoretical point of view, the two main claims of the paper (i.e. Theorems 4.2 and 4.3) are clear and I did not find any error in the proofs. About the experimental results: everything is convincing except for some comparisons, essentially in Tables 2-3  as I will detail below.

**Essential References Not Discussed:**

I don't think there are essential references not discusses, but I might be wrong.

**Experimental Designs Or Analyses:**

The experimental design is quite clear but it coud be improved. Consider Table 1, for instance. Since you simulate data from (high-dimensional) Gaussian distributions and you know the actual Wasserstein distance between them, you might calculate the absolute difference between the Wasserstein loss you come up with with your method, from each data sample, and the true one! Why to use the upper/lower bound of another quantity in this case? It is misleading. Moreover, and this is the weak point of the paper in my view, in Tables 2-3, you only compare your methods with Win and SCW_2B in terms of the lower bound BW_2^2-UVP. Now, a lower bound is a lower bound, what about the upper bound? Their UVP might be better than yours despite having a worse lower bound. Am-I missing something?

**Methods And Evaluation Criteria:**

Yes, most of the time the methods/evaluation criteria do make sense, although some comparisons with standard OT solver would be beneficial: barycenters similar to those in Figures 3-4 could be computed via standard discrete OT (e.g. in Python Optimal Transport) and it would be interesting to perform qualitative and quantitative comparisons with your method. E.g. what about the running times?

**Other Comments Or Suggestions:**

Just a couple of parentheses missing in the pseudo-codes such as the one at the very end of line 257.

**Other Strengths And Weaknesses:**

I liked very much this paper. It is well written, mathematics are elegant and the theoretical arguments are solid. I fell that Theorem 4.2 might  open the way to other approaches, adopting or not Normalizing Flows and, by the way, I consider that being able to solve OT problems (distance and barycenters) via standard SGD, in the primal, is an important advance.

As I sad, however I think that the experimental part could be improved in order to asses the interest of adopting this approach in place of alternatives.

**Questions For Authors:**

I would like to see this paper published at ICML. I will tentatively recommend is for "weak accept" but if you can do the one or two edits (Tables 1-2-3) that I suggested for the experimental part or alternatively show me that I am wrong about my concerns (it might be the case) I am ready to raise my note to "accept".

**Relation To Broader Scientific Literature:**

The relation with previous/related literature is correct, in my view.

**Theoretical Claims:**

I checked the correctness of proofs of part of Lemma A.3 and entirely revised proofs of Theorems 4.2 and 4.3. As I said everything looks correct at my eyes.

---

> ### Author Rebuttal · Authors · 2025-03-31
>
> Dear reviewer,
>
> thanks for taking the time to review our submission and for the kind appreciation of our contribution. We have carefully addressed all your comments below. According to the ICML review guidelines, we are not allowed to submit a revised manuscript at this point of the review process, but below we discuss in detail the edits we will make based on your suggestions in the camera-ready version of the paper, if accepted.
>
> **1. Some comparisons with standard OT solver would be beneficial: barycenters similar to those in Figures 3-4 could be computed via standard discrete OT (e.g. in Python Optimal Transport) and it would be interesting to perform qualitative and quantitative comparisons with your method. E.g. what about the running times?**
>
> The quantitative comparison between discrete OT and neural continuous OT methods has already been investigated extensively in the literature. The consensus is that discrete OT methods perform poorly in high dimensions, due to the required discretization of the input distributions. Specifically, performance degrades above $d=16$ (see, for instance, Figure 5 of (Fan et al, 2020) and Tables 1, 2, and 3 of (Korotin et al, 2021)), meaning that any discrete OT technique is assured to perform poorly on the MNIST 0/1 barycenter task shown in Figure 4 ($d=784$).
>
> On the other hand, discrete OT methods will certainly achieve good performance on the Swiss roll dataset (Figure 3) and decent computational times. Nevertheless, the point of presenting Figure 3 is not to stake a performance claim, but rather to verify visually that our conditional normalizing flow model performs well on highly non-linear distributions with complex support. The real performance evaluation is done on more high-dimensional problems in the subsequent sections.
>
> In conclusion, we think that a direct comparison with discrete OT methods in Figures 3 and 4 would not be meaningful.
>
> **2. Consider Table 1, for instance. Since you simulate data from (high-dimensional) Gaussian distributions and you know the actual Wasserstein distance between them, you might calculate the absolute difference between the Wasserstein loss you come up with with your method, from each data sample, and the true one! Why to use the upper/lower bound of another quantity in this case? It is misleading.**
>
> We completely agree with the reviewer. We re-ran Algorithm 1 on high-dimensional Gaussian data in the last few days and found that indeed our estimated Wasserstein distance converges to the true one very quickly. The plot for $d=64$ can be found here: https://pasteboard.co/jk8v0aZgFTom.png.
>
> This (and similar plots) will be added to the revised paper.
>
> **3. Moreover, and this is the weak point of the paper in my view, in Tables 2-3, you only compare your methods with Win and $SCW_2B$ in terms of the lower bound $BW_2^2-UVP$. Now, a lower bound is a lower bound, what about the upper bound? Their UVP might be better than yours despite having a worse lower bound.**
>
> This is a very important point, also raised by another reviewer. We entirely agree. Please see our answer to question 2 by reviewer u7iE.
>
> **4. I fell that Theorem 4.2 might open the way to other approaches, adopting or not Normalizing Flows.**
>
> This is definitely worth pursuing. We only mention that, since OT maps in Threorem 4.2 are ultimately computed by inverting the model, only invertible architectures are valid candidates for this approach.
>
> **5. Just a couple of parentheses missing in the pseudo-codes such as the one at the very end of line 257.**
>
> Thanks. We will add them in the revised paper.
>
> **References**
>
> Fan, Amirhossein, and Yongxin. "Scalable computations of wasserstein barycenter via input convex neural networks." arXiv:2007.04462 (2020).
>
> Korotin et al. "Continuous wasserstein-2 barycenter estimation without minimax optimization." arXiv:2102.01752 (2021).

---

### Official Review · Reviewer_u7iE · 2025-03-12

**Overall Recommendation:** 4

**Summary:**

This paper introduces a new way to compute optimal transport maps and Wasserstein barycenters using conditional normalizing flows.

**Claims And Evidence:**

By order of appearance:

1. Correctness of OT and barycenter problems: I have doubts on the barycenter derivation, as well as doubts on the construction of the loss, see theoretical claims section below.
2. Normalizing flows: There is some debate in the literature regarding universality, what papers are the authors referring to?
3. Yes, the method yields explicit maps, which is a great strength of this method.
4. Scaling: The presented evidence is convincing.


### Update after rebuttal: Main concern fixed

The barycenter proof was fixed, minor points see below.

**Essential References Not Discussed:**

Rectified Flows (and mini-batch flow matching) also try approximating the optimal transport between two distributions. This is not mentioned or compared to in the paper. The nature of the competing loss terms makes this challenging, of course.

References:

Pooladian, Aram-Alexandre, et al. "Multisample flow matching: Straightening flows with minibatch couplings." arXiv preprint arXiv:2304.14772 (2023).

Liu, Xingchao, Chengyue Gong, and Qiang Liu. "Flow straight and fast: Learning to generate and transfer data with rectified flow." arXiv preprint arXiv:2209.03003 (2022).


### Update after rebuttal: Mostly clarified

- The authors clarified the relation to Rectified Flows (Liu et al. 2022), and pointed me to the explanation in that paper.
- Regarding (Pooladian et al. 2023), the authors did not provide an explicit explanation, but I think the argument concerning rectified flows persists that there is no known cost that is minimized, and so the approaches are different

**Experimental Designs Or Analyses:**

I am unsure about the usefulness of the UVP metrics and hope for input from another reviewer.

**Methods And Evaluation Criteria:**

## OT Maps

Algorithm 1 is only evaluated on Gaussian data, which should be relatively easy to learn for a flow (e.g. Draxler et al. 2022 show that RealNVP flows converge exponentially fast to Gaussian data in terms of number of layers). I think estimating the Wasserstein distance for more complicated tasks would be a great add-on, e.g. pick any of the OT applications in the introduction. I think this is somewhat compensated by the Barycenter experiments, since OT estimation is a prerequisite for the barycenter computation.

Can you make a plot between the true Wasserstein distance and the distance estimated by the model?

## Barycenters

Why is there no L-UVP in Tables 2 and 3? The displayed loss is only a lower bound on the true metric.

Given that the rotation matrix in the 64-dimensional experiment in 5.2.4 is 2-dimensional, does the experiment really measure the scaling behavior?


## References

Draxler, Felix, Christoph Schnörr, and Ullrich Köthe. "Whitening convergence rate of coupling-based normalizing flows." Advances in Neural Information Processing Systems 35 (2022): 37241-37253.

### Update after rebuttal

Questions answered, experimental evidence still limited, but enough in my opinion.

**Other Comments Or Suggestions:**

l. 122 right: Add citation to McCann and Gangbo.

Please add links from the proofs of theorems in the main text to the appendix.

**Other Strengths And Weaknesses:**

The flow construction is innovative and easy to follow, nice work!

**Questions For Authors:**

Please take the criticism regarding the theory seriously. I will adjust my recommendation when the points are addressed.

## Extension ideas

Are there any of the real-world applications mentioned in the introduction feasible with this new method?

Wasserstein barycenters outside p=2? Relatedly, the "specialization" of Algorithm 1 to $p=2$ mentioned in the first paragraph of 4.3.2 affects only l. 202, right?

Is it possible to extend the algorithms to entropic OT? I guess this does not make sense since this produces non-unique couplings, right?


### Updates after rebuttal: Answers provided to all points

**Relation To Broader Scientific Literature:**

Optimal Transport is a principled mathematical framework for finding transport maps between distributions that adhere to external cost. It has a broad set of applications, but finding optimal transport maps in high dimensions remains an open challenge.

**Theoretical Claims:**

I did not check any statements in terms of where suitable maps exist, i.e. whether they are measurable. Instead I assumed that all densities are continuous wrt. Lebesgue measure.

## Theorem 4.2

Theorem 4.2 is convincing to me.

## Theorem 4.3 lacks proof, in my understanding

Let's first establish common ground, please confirm that I understand the proof of Theorem 4.3 correctly:

1. l. 707-715 show that h exists (since the Barycenter exists).
2. l. 715-716 argue that h establishes equality in Eq. 4 (since h + f(., s) are the solution of Theorem 4.2 to the OT between the Barycenter and each mu_s).
3. l. 717-722 show that h minimizes Eq. (l. 720) (if there was a better h, then it would not be the solution to the OT).
4. l. 722-724: Since h is the solution of that minimization, it is the average over the OT solution for each s (first statement of Thm 4.3).
5. Because p=2, we can separate the optimization tasks over dimensions.

However, I think the proof is missing the most important step: The map that satisfies Eq (l. 173) and Eq (l. 179) points to the barycenter has $h_\sharp \lambda = \bar \mu$. In addition, it seems that the second equation is not even used in Algorithm 2.

In other words: if one wants to train a model with these two as their loss, then one also has to show that the solution of this optimization is the barycenter.

This is my most important criticism, addressing it will increase my rating.


## Loss Tradeoff in Alg. 1 always introduces bias

I think that the argumentation regarding the annealing structure is incorrect and of different nature than simulated annealing. This is easy to see for the following counterexample: Let mu_1 and mu_2 be two Gaussian distributions with means $\pm 1$ and standard deviations $1$, and let the latent lambda also be a Gaussian distribution. Then the loss offers a closed-form solution which reveals that the learned means are off proportional to $\sqrt{\chi}$ (I might be wrong in the scaling).

In other words: The **optimal solution of the loss in Alg. 1 over all bijections will not be the optimal transport solution**. I would conjecture that it comes closer as $\chi \to 0$, but it is unclear how fast and what the tradeoff will be if presented arbitrary distributions.

Why different from annealing? In terms of suboptimal minima, I think that one can always move between different optima by sending infinitesimal packages of mass from one place to another -- so the loss barriers really are vanishing (and this effect seems to transfer to neural network parameterizations, see "mode connectivity" as per Garipov et al 2018, and Draxler et al 2018).

I think this will be an easy fix.

## Universality

I think there is a logical gap that the existing literature has not fully closed in terms of universality of normalizing flows. With the statement "normalizing flows, which are universal approximators for bijections and are thus the most natural generative models to employ in this context", the authors probably refer to the work by Teshima et al. 2020. However, as Koehler et al. 2021 and Draxler et al. 2024 point out, the underlying proofs use arbitrarily ill-conditioned networks (become arbitrarily ill-conditioned as the error decreases, e.g. Section 5.1 in Draxler et al. 2024).



## References

Garipov, Timur, et al. "Loss surfaces, mode connectivity, and fast ensembling of dnns." Advances in neural information processing systems 31 (2018).

Draxler, Felix, et al. "Essentially no barriers in neural network energy landscape." International conference on machine learning. PMLR, 2018.

Teshima, Takeshi, et al. "Coupling-based invertible neural networks are universal diffeomorphism approximators." Advances in Neural Information Processing Systems 33 (2020): 3362-3373.

Koehler, Frederic, Viraj Mehta, and Andrej Risteski. "Representational aspects of depth and conditioning in normalizing flows." International Conference on Machine Learning. PMLR, 2021.

Draxler, Felix, et al. "On the universality of volume-preserving and coupling-based normalizing flows." ICML (2024).


### Update after rebuttal

- Theorem 4.3 fixed.
- Bias: The authors argue that weight schedule on loss leads to correct solution, for which they provide additional evidence, which should be included in the paper and is good enough for now.
- Universality: Reached common ground on understanding of the literature

---

> ### Author Rebuttal · Authors · 2025-03-31
>
> Dear reviewer,
>
> thanks for the very constructive feedback! Apologies for our terse replies (due to character limit).
>
> **1. Can you make a plot between true and estimated $W_2$ distance?**
>
> See here for Gaussian OT ($d=64$): https://pasteboard.co/jk8v0aZgFTom.png. We will add such pictures.
>
> **2. Why is there no L-UVP in Tables 2-3?**
>
> The **L-UVP values for our model are available** (for $d=126$, L-UVP: 1.5% (gaussian), 5.5% (uniform)). We will add them to Tables 2-3. We left them out because Table 5 in (Korotin et al, 2022) does not report them. If appropriate, we can also report the L-UVP for SCWB and WIN (retrained by us).
>
> **3. The rotation matrix in 5.2.4 is only 2d.**
>
> Any high-dim. rotation is 2d in the right coordinates. Since the algo only sees realizations and has no "rotational" prior, the experiment does capture scaling behavior.
>
> **4. Theorem 4.3 lacks proof.**
>
> We wrote a clearer proof, which we sketch below.
>
> We know that
>
> $$\sum_s w_s W^2_2(\bar{\mu}, \mu_s) \le \sum_s w_s W^2_2(h_\\\#\lambda, \mu_s)  \le \sum_s w_s \\\| h - f_s \\\|^2_{L^2(\lambda)}$$
>
> for all $h:\mathbb{R}^d \to \mathbb{R}^d$ and all $f_s \in B(\lambda, \mu_s)$. Since $\bar{\mu} \in \mathcal{P}_{ac}(\mathbb{R}^d)$, Lem 4.1 and arguments in the proof of Thm 4.2 imply that there exist $h \in B(\lambda, \bar{\mu})$ and $f_s \in B(\lambda, \mu_s)$, such that
>
> $$ W^2_2(\bar{\mu}, \mu_s) = \\\| h - f_s \\\|^2_{L^2(\lambda)}$$
>
> and, therefore, achieve equality in the first chain of inequalities. This implies (see paper) that $h(Z) = \mathbb{E}[f(Z,S)|Z]$ and $f$ minimizes $\sum_{i=1}^d \mathbb{E}[\text{Var}(f_i(Z,S))]$.
>
> Assume that there is another function $\tilde{f}$ with $\tilde{f}(\cdot, s) \in B(\lambda, \mu_s)$ achieving minimal conditional variance. Then for $\tilde{h}(Z) = \mathbb{E}[\tilde{f}(Z,S)|Z]$ one has
>
> $$ \sum_s w_s W^2_2({\tilde{h}\\\#\lambda}, \mu_s) \le \sum_s w_s \\\| \tilde{h} - \tilde{f_s} \\\|^2_{L^2(\lambda)} = \sum_{i=1}^d \mathbb{E}[\text{Var}(\tilde{f}(Z,S))] = \sum_{i=1}^d \mathbb{E}[\text{Var}(f_i(Z,S))] = \sum_s w_s W^2_2(\bar{\mu}, \mu_s)$$
>
> Then by def of $\bar{\mu}$ we must have that $\tilde{h}_\\\# \lambda = \bar{\mu}$.
>
> **5. Eq (l. 179) is not used in Algorithm 2.**
>
> The L2 cost in Alg. 2 is exactly Eq (l. 179), where the conditional expectation is given by Eq (l. 173).
>
> **6. Loss Tradeoff in Alg. 1 always introduces bias.**
>
> Let us discuss the proposed example. If $\mu_s = \mathcal{N}(m_s, \sigma_s^2)$, for $s \in \{1, 2\}$, $\lambda = \mathcal{N}(0, 1)$ and $f(z, s|\theta) = \hat{\sigma}_s z + \hat{m}_s$, for $\theta = (\hat{m}_1, \hat{m}_2, \hat{\sigma}_1, \hat{\sigma}_2)$, then the model admits four MLEs: $ \theta^* \in \{(m_1, m_2, \sigma_1, \sigma_2), (m_1, m_2, -\sigma_1, \sigma_2), (m, m_2, \sigma_1, -\sigma_2), (m_1, m_2, -\sigma_1, -\sigma_2)\}$.
>
> The model OT map, $\hat{T}(x) = f(f^{-1}(x, 1), 2)$, is the right one only for two MLEs. In agreement with Thm 4.2, the right MLEs minimize $ L^2(z |\theta) = ((\hat{\sigma}_1 - \hat{\sigma}_2) z + (\hat{m}_1 - \hat{m}_2))^2$.
>
> In the reviewer's example ($\sigma_1 = \sigma_2 = 1$ and $m_1 \neq m_2$), at each $t$ in Alg. 1 we take a GD step towards the minimum of $ -\log(p(x,s|\theta)) + \zeta_t L^2(z |\theta)$:
>
> $$ \hat{m^*}_1(t) = \frac{m_1 + 2 \zeta_t \hat{m}_2}{1 + 2 \zeta_t}, \quad \hat{m^*}_2(t) = \frac{m_2 + 2 \zeta_t \hat{m}_1}{1 + 2 \zeta_t}.$$
>
> But $(\hat{m^*}_1(t), \hat{m^*}_2(t)) \to (m_1, m_2)$ as $\zeta_t \downarrow 0$, therefore at convergence there will be **no bias**. Empirically we do not observe any bias.
>
> **7. The argumentation regarding the annealing structure is incorrect.**
>
> Thanks for the insightful remarks. We do agree and we will remove any reference to annealing. We also saw empirically that minimizing the L2 cost with a likelihood regularizer works well, which shows that MLEs may lie on a connected manifold.
>
> **8. UATs for NFs use arbitrarily ill-conditioned networks.**
>
> This is a valid concern, but it applies to all UATs, which do not provide numerical stability guarantees. It is surely possible to construct distributions for which our networks are ill-conditioned. Empirically, the approach works well.
>
> **9. Other remarks**
>
> - We do have Alg. 1 results for uniform data (e.g. L-UVP=3.8%, BW-UVP=0.7%, for $d=126$). We will add them.
> - Rectified flows also use invertible nets, but do not compute *Wasserstein* OT maps nor barycenters.
> - Alg. 1 can be adjusted for generic $p>1$ in l.202. It's unclear if the same adjustment gives the $W_p$ barycenter in Alg. 2.
> - Any application in the paper intro is feasible for our method. See question 1 by reviewer U9wD.
> - Our method is not suitable for entropic OT, since we model Monge maps, not generic couplings.
>
> **References**
>
> - Korotin et al. "Wasserstein iterative networks for barycenter estimation." arXiv:2201.12245 2023
> - Liu, Chengyue, and Qiang "Flow straight and fast" arXiv:2209.03003 (2022).
> - Pooladian et al. "Multisample flow matching" arXiv:2304.14772 (2023).

---

> > ### Comment · Reviewer_u7iE · 2025-04-02
> >
> > Thanks for the helpful answers! I am listing the points I am still concerned about below:
> >
> > > More complicated OT estimation
> >
> > The authors did not react to this comment.
> >
> > > Plot between the true and the estimated W_2 distance
> >
> > Thanks for the plot, but I would rather suggest a plot (x=dimension y=OT distance, both ground truth and computed). It is good to see how quickly the OT converges to the true value. Is there any way of conservatively estimating the W_2, since learning a non-perfect distribution can lead to biased OT estimates?
> >
> > > L-UVP
> >
> > Yes, please report these numbers.
> >
> > > Theorem 4.3: New proof
> >
> > Looks correct to me, thanks for closing this gap in the logic.
> >
> > > Empirically, we do not observe any bias
> >
> > Looking at the plot provided by the authors in answer to my first question, I do observe a bias towards lower W_2, presumably at a cost to the accuracy. To test this, one would need to train a model without W_2 regularization and whether there is any performance drop by adding the regularization. Or how do the authors conclude that "empirically we do not observe any bias"?
> >
> > > UATs provide no numerical stability guarantees
> >
> > My point was that the UAT by Teshima et al. even requires arbitrary bad numerical stability for convergence [Koehler et al., Draxler et al.], *regardless of the target distribution*. I recommend rephrasing Contribution 2 and expanding on this point in the related work or theory section. Right now, my understanding is that it is misleading.
> >
> > Suggestion for what I mean:
> >
> > Contribution 2: ..., which are flexible bijections and thus the most natural generative model ...
> >
> > In 4.3.1 or related work: There are rigorous proofs that coupling-based normalizing flows are known to be universal approximators for bijections [Teshima], with the caveat that their construction relies on ill-conditioned networks [Koehler, Draxler]. [Draxler] present a well-conditioned flow, but they do not guarantee arbitrary bijections, but only approximating arbitrary distributions.
> >
> > > Rectified Flows do not compute Wasserstein OT
> >
> > Please see https://arxiv.org/pdf/2209.03003 Figure 3d and https://arxiv.org/pdf/2304.14772 Figure 6 -- there is a OT cost that is minimized because the Rectification step learns an optimized coupling. What am I missing?
> >
> > > More experiments
> >
> > I still think, like U9wD, 91Cy (more downstream tasks), and aK7B (more simple comparisons), that the experimental section could be improved.
> >
> >
> > I am looking forward to your answers!

---

> > > ### Author Response · Authors · 2025-04-09
> > >
> > > Dear reviewer,
> > >
> > > thanks again for your replies. We addressed your concerns below.
> > >
> > > **I think estimating the Wasserstein distance for more complicated tasks would be a great add-on, e.g. pick any of the OT applications in the introduction. [...] I still think [...] that the experimental section could be improved.**
> > >
> > > We have implemented a new numerical experiment related to a real-life application to fair regression, which we propose to add to the paper.
> > >
> > > Fair regression, in the sense of demographic parity, looks for a regression function $f(X,S)$ that minimizes the cost $\mathbb{E}[\\\|Y - f(X,S)\\\|^2]$, such that $f(X,S)$ is independent of $S$. The solution to this constrained optimization is precisely the Wasserstein barycenter of the conditional distributions of $Y$ given $S$ (see (Chzhen et al, 2020)).
> > >
> > > We work on the benchmark dataset "Communities and Crime'' and we regress the target variables "percentage of officers assigned to drug units'' and "total number of violent crimes per 100K popuation'' on 127 socio-economic features ($X$) and one sensitive feature ($S$), the "percentage of population that is African-American", with range $[0,1]$.
> > >
> > > An "unfair" regression leads to strong correlation between, for instance, the first predicted variable and the sensitive feature (Pearson: 0.43, Spearman: 0.47, Kendall: 0.32), while our fair regression achieves almost perfect uncorrelatedness (Pearson: 0.0003 (p-value: 0.99), Spearman: 0.0058 (p-value: 0.80), Kendall: 0.0039 (p-value: 0.80)). In all cases, we fail to reject the null hypothesis of no association.
> > >
> > > This experiment requires computing the barycenter of $100$ input distributions (the cardinality of the range of $S$, rounded up to the nearest % point), which would be computationally challenging for any other numerical method. We think that this experiment exemplifies the relevance of our method for real-life applications.
> > >
> > > **I would rather suggest a plot (x=dimension y=OT distance, both ground truth and computed).**
> > >
> > > Thanks for the nice suggestion. We made the plot for $d=16, 32, 64$: https://pasteboard.co/EJUgnxvRkesi.png. On the y-axis, we report a box-plot of the (signed) relative error of the estimated Wasserstein distance for 10 random initializations of our model.
> > >
> > > **Is there any way of conservatively estimating the W2?**
> > >
> > > Yes, this is exactly the idea behind the BW-UVP metric. The $W_2$ distance between two distributions is always lower-bounded by the $W_2$ between two Gaussians with the their respective means and covariances (see (Dawson and Landau, 1982)). This gives a conservative estimate under assumption of normality.
> > >
> > > **How do the authors conclude that ``empirically we do not observe any bias''.**
> > >
> > > In all experiments, we checked for bias:
> > > - by visual inspection of the marginals (as done in our Figure 1),
> > > - by comparing the true and estimated means (resp. covariances) in terms of Euclidean (resp. Frobenius) norm,
> > > - by computing the L-UVP and BW-UVP metrics, which are just (normalized) upper/lower bounds on the $W_2$ distance between the estimated and the true target measure, and therefore quantify bias.
> > >
> > > We think that the new figure (linked above), together with the theoretical considerations in reply to Question 6 in our previous message, show strong evidence that our model is free of bias.
> > >
> > > **The UAT by Teshima et al. even requires arbitrary bad numerical stability for convergence [Koehler et al., Draxler et al.], regardless of the target distribution.**
> > >
> > > Thanks for expanding on this point, we fully understand now. We will add a discussion of these issues in our Contribution 2 and discussion of related works, exactly as you suggest. Thanks again for pointing out this gap.
> > >
> > > **Please see (Liu et al, 2022) Fig 3d and arXiv:2304.14772 Fig 6 -- there is a OT cost that is minimized because the Rectification step learns an optimized coupling. What am I missing?**
> > >
> > > OT is concerned with finding optimal couplings that minimize a pre-specified transport cost $c$. In our paper we deal with the Wasserstein OT problem, where the cost is $c(x,y) = |x-y|^p$, for $p>1$. The rectified flow, instead, aims at minimizing a path functional for an SDE with given initial and terminal marginals (see Eq (1) in (Liu et al, 2022)).
> > >
> > > The relationship between the two is explained in Sec 3.4 of (Liu et al, 2022), where they show that rectified couplings (i.e. couplings that minimize the rectified flow):
> > > - are not optimal under any cost $c$ (in particular they do *not* correspond to $W_2$ OT couplings),
> > > - in general provide only a lower-bound for the OT cost with respect to any convex cost $c$ (Thm 3.8), as they verify empirically in their Fig 3(d).
> > >
> > > **References**
> > >
> > > - Dowson, Landau. "The Fréchet distance between multivariate normal distributions." Journal of multivariate analysis 12.3 (1982): 450-455.
> > > - Chzhen et al. "Fair regression with wasserstein barycenters." NeurIPS 33 (2020): 7321-7331.
> > > - Liu, Chengyue, and Liu. "Flow straight and fast" arXiv:2209.03003 (2022).

---

### Official Review · Reviewer_91Cy · 2025-03-13

**Overall Recommendation:** 2

**Summary:**

The authors propose a new method for finding Wasserstein-2 barycenter via Conditional Normalizing Flows (CNF) as well as computation of Optimal Transport (OT) maps from input distributions to the barycenter. The key advantage of the method is minimization of the primal OT problem by invertible pushforward bijections, avoiding adversarial bi or tri-level optimization problems present  in the previous methods. They demonstrate performance of the method for computing Wasserstein-2 barycenter for Swiss Roll dataset, Gaussians and high-dimensional case for MNIST dataset.

## **Update after rebuttal.**

I have carefully read the authors' responses and would like to raise a few concerns:
- (minor) Upon re-reading the manuscript, I noticed that the paper (Korotin et al., 2019) is incorrectly cited as representing the WIN approach in several places (lines 142, 385, 394). I believe the authors intended to reference (Korotin et al., 2022). This appears to be a typo and should be corrected, as it may confuse readers.
-  I am *confused* by the fact that you do not use the time of rebuttal to perform the comparison with some of the non-generative approaches which I pointed to (Kolesov 2024a,b). There are not so many papers on continuous OT barycenter estimation and even less of them are generative (in your classification). Thus, it seems for me that the comparison with the best non-generative approaches in the appropriate experimental setups is (1) meaningful and (2) not so time-consuming. It is even more confusing since your section 4.3.2 is dedicated to the similar idea of computing the barycenter using the conditional normalizing flows, which allow for computation of OT maps between input distributions and barycenter (i.e., finding barycenter points from samples of input distributions), and the experimental section 5.2 tests this conditional model.  At least in experiment with *location-scatter Gaussian* data the comparison is possible and **meaningful**. See my next point for details.
- I previously referred to the L$_2$-UVP values in Table 1 for the Gaussian experiment, which was incorrect. I actually meant the experiment with location-scatter Gaussian data. As another reviewer noted, the L$_2$-UVP comparison is missing for this case. I believe that, in this experiment, comparing against both generative and non-generative methods would be valuable for situating your approach within the current landscape of continuous barycenter solvers.

- Besides, I am confused by the overall practical validity of the proposed approach, because (1) it works only with the quadratic cost, (2) was not tested on any practical experimental setups considered in SC$\mathbb{W}_2$B (Fan et al, 2021) or WIN (Korotin et al., 2022) papers or any others. For example, the authors do not test their approach on "Ave, celeba!" benchmark dataset considered in WIN paper. It is strange because both competitors SC$\mathbb{W}_2$B and WIN were tested here (see section 6.1 of WIN's paper). Instead, the authors argue that such experiments are not meaningful. However, given that both SC$\mathbb{W}_2$B and WIN were evaluated on this benchmark, omitting it weakens the comparison. (Actually, here it is not necessary to use your conditional model - the comparison of the learned barycenters is enough to measure the performance of your approach.)

Overall, I remain unconvinced that the current contribution is substantial enough to warrant publication at this conference.

**References.**

Fan, J., Taghvaei, A., and Chen, Y. Scalable computations of wasserstein barycenter via input convex neural networks. arXiv preprint arXiv:2007.04462, 2020.

Korotin, A., Egiazarian, V., Asadulaev, A., Safin, A., and Burnaev, E. Wasserstein-2 generative networks. arXiv preprint arXiv:1909.13082, 2019.

Korotin, A., Egiazarian, V., Li, L., and Burnaev, E. Wasserstein iterative networks for barycenter estimation. Advances in Neural Information Processing Systems, 35: 15672–15686, 2022.

Kolesov, A., Mokrov, P., Udovichenko, I., Gazdieva, M., Pammer, G., Burnaev, E., and Korotin, A. Estimating barycenters of distributions with neural optimal transport. arXiv preprint arXiv:2402.03828, 2024a.

Kolesov, A., Mokrov, P., Udovichenko, I., Gazdieva, M., Pammer, G., Kratsios, A., ... & Korotin, A. (2024b). Energy-guided continuous entropic barycenter estimation for general costs. Advances in Neural Information Processing Systems, 37, 107513-107546.

**Claims And Evidence:**

Most of the claims are supported by clear evidence. However, some points of the submission remain unclear from me:

- **Limitations of the approach.** The authors do not discuss the limitations of the proposed method. It is an important point since normalizing flows are known to suffer from different drawbacks (e.g., constraints on the architecture, high computational expense) and the proposed approach should have inherited the same issues which is not directly stated in the text;

- **Scalability.** It is stated that the approach is applicable for high-dimensional datasets, however, the only high-dimensional experiment considered un the paper corresponds to the computation of barycenter for MNIST images which have the dimension 28x28. It is not clear how the method behaves of experiments with higher dimensions, see, e.g., “Ave, CelebA” experiment from (Kolesov et al, 2024a).

- **Limited comparison.** In the considered experimental setups, the authors do not compare their approach with the recent approaches for barycenter estimation, e.g., (Kolesov 2024a,b). As it is evident from the results reported in this papers, their performance is much better than that of WIN (Korotin et al., 2022), thus, the comparison with them is crucial for understanding the performance of the proposed approach. Since it is possible to retrieve the OT maps using your approach, such a comparison seems to be valid.

**Essential References Not Discussed:**

The authors have cited most of the related literature.

**Experimental Designs Or Analyses:**

Experimental results are overall valid. However, I have several concerns which I have written in 'claims and evidence' section.

**Methods And Evaluation Criteria:**

The proposed method and evaluation criteria make sense for the considered problem.

**Other Comments Or Suggestions:**

- The authors claim that (Kolesov et al, 2024) does not provide a generative model of barycenter. However, it is not indeed true. StyleGAN is a manifold-constrained generator in the aforementioned method and it directly samples from barycenter.

**Other Strengths And Weaknesses:**

**Strengths**:

- This method proposes non-adversarial optimization problem for finding Wasserstein-2 barycenter;
- The authors  offer new intuitive reformulation of the Wasserstein barycenter as expected conditional variance of pushforward bijective transformations;
- The authors develop new generative model that is able to sample directly from barycenter.

**Weaknesses**:

- The authors do not mention limitations of their approach in the main text. For example, since normalizing flows have drawbacks as constraints of considered transformations , high computational expense and instability, these  shortcomings are being moved to the proposed method.
- While the dimensionality of tasks grows, the conditional variance grows too. As a consequence of this, the variance of the loss function grows too and it leads to another point of the method’s instability.
- The comparison with other approaches has limitations, see previous sections.

**Questions For Authors:**

- The authors claim that their approach performs well on high-dimensional data taking MNIST dataset as  a high-dimensional experiment. Is the developed method appropriate for the “Ave, CelebA” experiment (See p.9 from (Kolesov et al, 2024) )?
- The authors provide comparisons with WIN and SCWB. However, (Kolesov et al, 2024) is also suitable for the finding barycenter in MNIST dataset experiment, could you compare with it?
- What about convergence time of the proposed method? One would like to see comparisons of convergence times for the proposed approach with SCWB, WIN and  (Kolesov et al, 2024).

**References.**

Kolesov, A., Mokrov, P., Udovichenko, I., Gazdieva, M., Pammer, G., Burnaev, E., and Korotin, A. Estimating barycenters of distributions with neural optimal transport. arXiv preprint arXiv:2402.03828, 2024.

**Relation To Broader Scientific Literature:**

The paper proposes a new approach for finding a barycenter of distributions which avoids min-max optimization objective by using the normalizing flows. As far as I know, the usage of normalizing flows in this context is quite novel.

**Theoretical Claims:**

Yes, I skimmed through the theoretical results and their proofs and do not find any issues.

---

> ### Author Rebuttal · Authors · 2025-03-31
>
> Dear reviewer,
>
> please find our replies below.
>
> **1. The authors do not discuss limitations of the proposed method.**
>
> We agree that discussing the limitations is very important. We propose to add discussions of the following limitations:
> 1. Alg. 2 is suitable only for Wasserstein-2 barycenters.
> 2. Training might be computationally intensive on image datasets larger than the ones we test ($\gg$1000 dimensions).
>
> **2. Normalizing flows (NFs) have known drawbacks (e.g., constraints, high computational cost, instability).**
>
> In our experience NFs are stable and reliable generative models, as our results show. They require large latent spaces, due to the **bijectivity constraint**, but the multi-scale architecture makes training in high dimensions unproblematic (see App. D). Training is **stable** with minimal fine-tuning (e.g. only lower learning rate on non-Gaussian data, see App. D). Our method did not require more **compute** than competitors and can run on a single GPU with 12GB. For training times, see question 7.
>
> **3. Is the method appropriate for “Ave, CelebA”?**
>
> Yes. Indeed, the Glow model (which we build on in Sec 5.2.3) was trained on the CelebA-HQ dataset with great success (Kingma e al., 2018), but it required one week of training on 40 GPUs (https://github.com/openai/glow/issues/37). We lack such computational resources. We will mention this in the method's limitations (see question 1).
>
> We claim that our experimental set-up is sufficient to prove the **scability of our method**, because the MNIST dataset ($d=784$) is high-dimensional enough **for all real-life applications** (see paper references): style transfer ($d=500$), color translation ($d=3$), clustering in Wasserstein space ($d=10-200$), fairness: ($d=1-200$).
>
> Notice that "Ave, CelebA'' is not related to any real-life task (see question 2 by reviewer U9wD).
>
> **4. The authors do not compare their approach with the recent approaches for barycenter estimation, e.g., (Kolesov 2024a,b), which have much better performance than that of WIN (Korotin et al., 2022).**
>
> Generative methods solve a strictly harder problem than OT map estimation. To keep the comparison fair, we compared **generative models only** (see question 5).
>
> Furthermore, the StyleGAN methods in (Kolesov 2024,a,b) solve an OT problem in the StyleGAN latent space with non-quadratic cost. Since none of the methods in our paper supports generic costs, a **comparison is impossible**.
>
> In terms of model evaluation, nothing would be gained by adding (Kolesov 2024a,b), because WIN has a very similar performance (see Table 3 in (Kolesov 2024a) and our Tables 2-3 for location-scatter data and Fig 5 (a, b) in (Kolesov 2024b) for MNIST data).
>
> **5. StyleGAN is a manifold-constrained generator in (Kolesov et al, 2024a)  and it directly samples from barycenter.**
>
> We explain why that's not the case. Barycenter generative models "output a learned barycenter distribution, which can be directly sampled, while non-generative models are limited to transporting existing samples from the input distributions" (see Sec 3). The pre-trained StyleGAN $G$ in (Kolesov et al, 2024a,b) (which is an *input* of the method, not an output) is **not** the barycenter (see their Fig 3). A barycenter sample can be obtained only by pushing an input distribution sample $x_k$ through the OT map $T_{k, \phi}(x_k, z)$ (see their Fig 1 (a,b)). In our method, instead, barycenter samples are obtained by pushing latent samples through the map $h$ (no input distribution query needed).
>
> **6. Could you compare with (Kolesov et al, 2024a) on the MNIST dataset?**
>
> See questions 4 and 5, above.
>
> **7. What about convergence time of the proposed method?**
>
> Training time on Gaussian location-scatter data ($d=128$) are: 114m56s (SCWB), 93m28s (WIN), 42m 3s (ours). All methods converge in approximately 10-20 minutes, longer times needed only for top-notch performance. On MNIST our method converges in 30 minutes. We will provide full information on training times.
>
> **8. While the dimensionality of tasks grows, the conditional variance grows too [...] and it leads to [...] instability.**
>
> The conditional variance scales linearly in the number of dimensions $d$. Let $f(Z,s)$ have mean $m_s$ and covariance $\Sigma_s$ and denote $\bar{m} := \sum_{s \in \mathcal{S}} w_s m_s$, then by the law of total variance:
> $$\sum_{i=1}^d \mathbb{E}\left[ \text{Var}(f_i(Z,S) | Z) \right] \le \sum_{i=1}^d \text{Var}(f_i(Z,S)) = \sum_{s \in \mathcal{S}} w_s \rm{Tr}(\Sigma_s) + \sum_{s \in \mathcal{S}} w_s \| m_s - \bar{m}\|^2 = O(d)$$
> This growth rate is the same as multivariate regression and is unproblematic.
>
> **References**
> - Kingma, Prafulla. "Glow: Generative flow with invertible 1x1 convolutions." NeurIPS 31 (2018).
> - Kolesov et al. "Estimating barycenters of distributions with neural OT." arXiv:2402.03828 (2024a).
> - Kolesov et al. "Energy-guided continuous entropic barycenter estimation for general costs." NeurIPS 37 (2024b): 107513-107546.

---

> > ### Comment · Reviewer_91Cy · 2025-04-03
> >
> > Thank you for your answers. There are several claims in your answer which I can not agree with, see below.
> >
> > > Notice that "Ave, CelebA'' is not related to any real-life task (see question 2 by reviewer U9wD).
> >
> > In your answer to reviewer U9wD, you explain that computing the barycenter of images using the quadratic cost is not meaningful. It is a known issue which motivates the researches to consider more elaborated cost functions and highlights the limitation of your approach. The aspect that your approach is limited to the quadratic cost should be mentioned in limitations section. Of course, "Ave, Celeba" experiment as well as all experiments considered in your paper are not related to real-world life but they are used to highlight the properties of your approach.
> >
> > > WIN has a very similar performance to (Kolesov 2024a,b)
> >
> > It is not true - in Table 3 of (Kolesov 2024a), WIN provides **ten times worse** performance than their approach in moderate dimensions (d=64) and much smaller convergence time.
> >
> > > Comparison with (Kolesov 2024a,b) is not relevant
> >
> > I can not agree on this point. To start with, I remain skeptical regarding the applicability of *generative* barycenters to real-world tasks. I do not see any sufficient explanations in the text of your paper as well as in the answers to reviewers. Meanwhile, in your experiments, you consider only toy experiment (Gaussians, Swiss roll, etc.) and MNIST one where the aim was to generate barycenter *from the samples of the input distribution*. I understand where this kind of task might appear in real-life, but I do not understand where the generative task might appear. I think you should add more discussion on its applicability in actuarial sciences which you mention.
> >
> > And I think that since you explore the ability of your approach to perform the translation from the input samples to barycenter you should perform the comparison with the SOTA solvers (Kolesov a,b). I do not see any obstacles to do it in MNIST experiment during the remaining rebuttal time.
> >
> > You can also easily perform comparison in Gaussians experiment where all of the approaches (generative and non-generative) are applicable. I do not see the details on the Gaussian experiment (you do not specify the weights of barycenters), but I guess that you took the same setup as was used in (Korotin et al., 2022) and also in Kolesov et al., 2024a. Then according to their results in Table 3 and $\mathcal{L}_2$-UVP values reported for your approach, your method provides worse results than WIN and (Kolesov a, b)  in all dimensions except for dimension D=2. It poses even more questions about the applicability of your approach.
> >
> > Looking forward for your replies!

---

> > > ### Author Response · Authors · 2025-04-09
> > >
> > > Dear reviewer,
> > >
> > > thanks for your feedback. Please, find our replies below.
> > >
> > > **The aspect that your approach is limited to the quadratic cost should be mentioned in limitations section.**
> > >
> > > Yes, thanks for the suggestion. We agree to mention this limitation.
> > >
> > > **In Table 3 of (Kolesov 2024a), WIN provides ten times worse performance than their approach in moderate dimensions (d=64) and much smaller convergence time.**
> > >
> > > Thanks for pointing out that the performance of WIN is not as good as (Kolesov 2024a). Both papers are very interesting. But since our method is generative, it makes more sense to compare our method to WIN.
> > >
> > > **I remain skeptical regarding the applicability of generative barycenters to real-world tasks. I do not see any sufficient explanations in the text of your paper as well as in the answers to reviewers. [...] I think you should add more discussion on its applicability in actuarial sciences which you mention.**
> > >
> > > Thanks for your suggestion. We completely agree and will add a more in-depth discussion of the benefits of generative models for Wasserstein barycenters. For many applications, such benefits have been already been highlighted in their respective papers (see our references in the introduction). We report a few of them here:
> > > - Shape/image interpolation: the density of the generative model is already the interpolating shape and does not require surface reconstruction from point clouds.
> > > - Style transfer: the barycenter generative model can generate previously unseen samples (e.g. new MNIST digits in a given style), as opposed to just transporting already existing ones (MNIST digits are, after all, finitely many).
> > > - Fair regression: the barycenter generative model effectively performs distributional regression for the fair premium, which allows, for instance, the construction of confidence intervals, not just point estimates.
> > > - Generative models make it possible to estimate any statistic of the barycenter distribution arbitrarily well (as useful, for instance, in subset posterior estimation). It also allows to transform the distribution further (e.g. it can be shifted so that it has zero mean, which is useful in actuarial applications).
> > >
> > > **And I think that since you explore the ability of your approach to perform the translation from the input samples to barycenter you should perform the comparison with the SOTA solvers (Kolesov a,b). I do not see any obstacles to do it in MNIST experiment during the remaining rebuttal time. You can also easily perform comparison in Gaussians experiment where all of the approaches (generative and non-generative) are applicable.**
> > >
> > > The papers (Kolesov 2024a,b) are great, but, as we mentioned in Questions 4 and 5, their methods are not directly comparable with ours because:
> > > - they are not generative: they learn the OT maps from the barycenter to the input distributions, but they don't train a generative model of the barycenter,
> > > - the StyleGAN-based implementation solves a manifold-constrained OT problem with non-quadratic cost, which is not supported by our method, nor by the other baseline models.
> > >
> > > **Then according to their results in Table 3 and L2-UVP values reported for your approach, your method provides worse results than WIN and (Kolesov a, b) in all dimensions except for dimension D=2. It poses even more questions about the applicability of your approach.**
> > >
> > > Thanks for your feedback. But please, notice that these are two different numerical experiments that cannot be compared. Our Table 1 reports results of an OT problem transporting a source distribution $\mu$ to a target distribution $\nu$, while Table 3 in (Kolesov 2024a) refers to a barycenter problem (input measures $(\mu_s)$ are transported into their barycenter $\bar{\mu}$).
> > >
> > > **References**
> > > - Kolesov et al. "Estimating barycenters of distributions with neural OT." arXiv:2402.03828 (2024a).
> > > - Kolesov et al. "Energy-guided continuous entropic barycenter estimation for general costs." NeurIPS 37 (2024b): 107513-107546.

---

### Official Review · Reviewer_U9wD · 2025-03-13

**Overall Recommendation:** 4

**Summary:**

The paper proposes a new algorithm to compute the Wasserstein barycenter of a set of distributions and, by extension, the optimal transport between any pair of the given distributions. The method rests on a new representation of the barycenter objective in terms of conditional normalizing flows. Since this objective can be readily approximated by empirical averages and variances, the resulting algorithm is very simple and elegant. Experiments illustrate the algorithm's outcomes on the Swiss roll and the MNIST datasets and demonstrate its superiority over the competition in systematic comparisons.

**Claims And Evidence:**

The paper derives and proves the new variant of the objective (theorem 4.3). Since the proof short-cuts some steps by citing results from the literature, its correctness it not readily apparent. The experiments are well designed and support the superiority claim.

**Essential References Not Discussed:**

None found.

**Experimental Designs Or Analyses:**

See above.

**Methods And Evaluation Criteria:**

The experiments adapt an evaluation protocol with analytic ground truth from the literature. This makes results directly comparable and supports the superiority claim.

**Other Comments Or Suggestions:**

The authors might want to consider putting the proofs of theorems 4.2 and 4.3 in the main paper. They are not very long, the paper seems to have substantial unused white space, and it would make the presentation more self-contained.

Minor points and typos:
* The formal definition of the L_p(lambda)-norm is missing. This makes theorem 4.2 hard to understand.

* It is never specified over which distributions expectations and variances are taken. This makes theorem 4.3 hard to understand.

* A closing bracket is missing in the change-of-variables formulas in algorithms 1 and 2.

* Line 205: "s may be univariate discrete". Do you mean "univariate continuous"?

* Section 5.2.1: The weights w_s are not specified.

**Other Strengths And Weaknesses:**

The authors note -- in accordance to earlier findings -- that samples from the MNIST barycenter look like a superposition of 10 digits with the same style. It should be discussed what this means: Is the barycenter a useful notion of the "average" of a set of given distributions? Does it have appealing properties beyond minimizing the Wasserstein distance? If yes, are these properties useful to simplify or enable downstream tasks? If no, what's the point of computing barycenters?

**Questions For Authors:**

Can you demonstrate the usefulness of Wasserstein barycenters for downstream tasks?

**Relation To Broader Scientific Literature:**

The experiments only include two alternative methods. I do not know if there are others that should have been included.

**Theoretical Claims:**

See above.

---

> ### Author Rebuttal · Authors · 2025-03-31
>
> Dear reviewer,
>
> we have carefully addressed all your comments below. According to the ICML review guidelines, we are not allowed to submit a revised manuscript at this point of the review process, but below we discuss in detail the edits we will make based on your suggestions in the camera-ready version of the paper, if accepted.
>
> **1. Can you demonstrate the usefulness of Wasserstein barycenters for downstream tasks?**
>
> Wasserstein barycenters have found application in many downstream tasks, such as shape interpolation, image interpolation, color translation, style translation, Bayesian subset posterior estimation, clustering in Wasserstein space, and fairness (for references for each application, see the introduction of the paper).
>
> Our interest in real-life applications to fairness in insurance prompted us to develop a method with good scalability properties in the number of input distributions, which is absent in the existing literature and one of the main contributions of our paper. Since the method is of independent interest for the broader Wasserstein barycenter community, in this paper we focus on introducing it and showcasing its competitive performance on well-established benchmarks, leaving its application to real-life actuarial datasets for upcoming work.
>
> **2. Samples from the MNIST barycenter look like a superposition of 10 digits with the same style. Is the Wasserstein barycenter a useful notion?**
>
> The notion of Wasserstein barycenter is in general non-trivial. For instance, the Wasserstein-2 barycenter of two Dirac measures $\delta_x$ and $\delta_y$ (for $x,y \in \mathbb{R}^d$) is not the measure $\frac{1}{2} \delta_x + \frac{1}{2} \delta_y$, but rather the Dirac measure centered at their midpoint, i.e. $\delta_{\frac{x+y}{2}}$. Thanks to this property, Wasserstein barycenters can provide very natural-looking averages of 2d/3d images, *when each image is modelled as an input distribution in 2d/3d space*, as can be seen in Figures 1 and 7 of (Solomon et al, 2015) or Figures 1 and 2 of (Cuturi et al, 2014).
>
> Recent works, instead, focus on image datasets in which *input distributions are not themselves images, but rather empirical distributions of images* (as in the MNIST barycenter task in our Section 5.2.3). While these benchmarks may be useful to showcase model scalability, they are of very limited practical interest (the barycenter images are just pixel-wise averages of input images) and do not correspond to any meaningful image interpolation task. We agree with the reviewer and acknowledge that there is a pressing need for the Wasserstein barycenter community to develop more meaningful benchmark tasks in high dimensions, possibly moving away from image data. We hope our future work on actuarial datasets will contribute to this.
>
> **3. It is never specified over which distributions expectations and variances are taken. This makes theorem 4.3 hard to understand.**
>
> We follow the standard convention of assuming that there is an underlying probability space on which all random variables are defined (this can be done without loss of generality, see Lemma 8.16 of (Kallenberg, 2002)). All expectations are then well-defined Lebesgue integrals with respect to this probability measure. We will add this assumption explicitly in the paper.
>
> **4. Line 205: "s may be univariate discrete". Do you mean "univariate continuous"?**
>
> By "univariate discrete'', we actually meant an $\mathbb{R}$-valued random variable taking finitely many values (such as the discrete uniform distribution). To avoid misunderstandings, we will edit the text in l.205 to read "[...] the conditioning variable $s \in \mathcal{S}$ may be $\mathbb{R}$-valued and taking finitely many values (as in Section 5.2.4, where $\mathcal{S}$ is the uniform grid on $[0, \pi]$ with $n$ points) or one-hot encoded (as in all other numerical experiments).''
>
> **5. The authors might want to consider putting the proofs of theorems 4.2 and 4.3 in the main paper. They are not very long, the paper seems to have substantial unused white space, and it would make the presentation more self-contained.**
>
> This is a good suggestion. We will do this, provided we can respect the ICML page limit for the revised paper.
>
> Thanks also for reporting the typos, missing definitions and missing parameter values. We will correct them all in the revised paper.
>
> **References**
>
> - Solomon, Justin, et al. "Convolutional wasserstein distances: Efficient optimal transportation on geometric domains." ACM Transactions on Graphics (ToG) 34.4 (2015): 1-11.
> - Cuturi, Marco, and Arnaud Doucet. "Fast computation of Wasserstein barycenters." International conference on machine learning. PMLR, 2014.
> - Kallenberg, Olav. Foundations of Modern Probability. Springer Science \& Business Media, 2002.

---

> > ### Comment · Reviewer_U9wD · 2025-04-02
> >
> > > Answer 2: While these benchmarks may be useful to showcase model scalability, they are of very limited practical interest (the barycenter images are just pixel-wise averages of input images) and do not correspond to any meaningful image interpolation task.
> >
> > For better or worse, the shortcomings of barycenters for the image dataset (superposition of the classes) are readily visible to the human eye. For more abstract distributions (e.g. data from medicine or biology) these shortcomings may be much less apparent (and thus go undetected), but still detrimental for the barycenter to be useful for downstream tasks. I'm thus not yet fully convinced that barycenters are really the right definition of an "average" between a set of distributions, despite their popularity as a research topic in its own right. It would be good to add at least some discussion of this problem to the paper.
> >
> > > Answer 3: We follow the standard convention of assuming that there is an underlying probability space on which all random variables are defined,
> >
> > I know that conventions like this are very common in mathematics (similar to dropping parentheses because the "correct grouping of terms is obvious" -- no, it is not). As a computer scientist, I find these conventions really stupid. Being explicit immensely helps readability of the equations. Computer science learned the hard way that readability is much more important than brevity! I thus tend to insist that you explicitly write out over which distributions the expectations are taken (typically as a subscript to the E symbol).

---

> > > ### Author Response · Authors · 2025-04-09
> > >
> > > Dear reviewer,
> > >
> > > thanks for your replies.
> > >
> > > **I'm thus not yet fully convinced that barycenters are really the right definition of an "average" between a set of distributions, despite their popularity as a research topic in its own right. It would be good to add at least some discussion of this problem to the paper.**
> > >
> > > Thanks for your suggestion. We agree that Wasserstein barycenters are only one possible notion of average between distributions and that other notions could also make sense. However, there are several contexts in which Wasserstein barycenters have been proven to be optimal, as in the following cases:
> > > - Theorem 2.3 of the NeurIPS paper (Chzhen et al, 2020) shows that optimal fair predictors are given by Wasserstein barycenters,
> > > - Theorem 4 of the JMLR paper (Srivastava et al, 2018) shows that the barycenter of subset posteriors converges to the true posterior distribution.
> > >
> > > We think that these results are strong evidence for the usefulness of Wasserstein barycenters. We agree with your suggestion and we will add more motivation of the relevance of Wasserstein barycenters in our paper.
> > >
> > > **I know that conventions like this are very common in mathematics (similar to dropping parentheses because the "correct grouping of terms is obvious" -- no, it is not). As a computer scientist, I find these conventions really stupid. Being explicit immensely helps readability of the equations. Computer science learned the hard way that readability is much more important than brevity! I thus tend to insist that you explicitly write out over which distributions the expectations are taken (typically as a subscript to the E symbol).**
> > >
> > > We agree that clarity is important. We will clarify the statement and proof of Theorem 4.3.
> > >
> > > **References**
> > >
> > > Chzhen et al. "Fair regression with wasserstein barycenters." Advances in Neural Information Processing Systems 33 (2020): 7321-7331.
> > >
> > > Srivastava et al. "Scalable Bayes via barycenter in Wasserstein space." Journal of Machine Learning Research 19.8 (2018): 1-35.

---

### Decision · Program_Chairs · 2025-05-01

**Decision:**

Accept (poster)

**Comment:**

In this study, the authors propose a conditional normalizing flow method to compute optimal transport maps and Wasserstein barycenters. All four reviewers, including AC, agree that the proposed method is interesting and solid in theory. However, AC and Reviewer 91Cy require the authors to polish the experimental part and introduce the implementation details clearly. In summary, AC tends to accept this work.